# Kinesin-4 KIF21B limits microtubule growth to allow rapid centrosome polarization in T cells

Peter Jan Hooikaas[1†], Hugo GJ Damstra[1†], Oane J Gros[1], Wilhelmina E van Riel[1‡], Maud Martin[1§], Yesper TH Smits[2], Jorg van Loosdregt[2], Lukas C Kapitein[1], Florian Berger[1*], Anna Akhmanova[1*]

[1]Cell Biology, Neurobiology and Biophysics, Department of Biology, Faculty of Science, Utrecht University, Utrecht, Netherlands; [2]Center for Translational Immunology, University Medical Center Utrecht, Utrecht University, Utrecht, Netherlands

**Abstract** When a T cell and an antigen-presenting cell form an immunological synapse, rapid dynein-driven translocation of the centrosome toward the contact site leads to reorganization of microtubules and associated organelles. Currently, little is known about how the regulation of microtubule dynamics contributes to this process. Here, we show that the knockout of KIF21B, a kinesin-4 linked to autoimmune disorders, causes microtubule overgrowth and perturbs centrosome translocation. KIF21B restricts microtubule length by inducing microtubule pausing typically followed by catastrophe. Catastrophe induction with vinblastine prevented microtubule overgrowth and was sufficient to rescue centrosome polarization in KIF21B-knockout cells. Biophysical simulations showed that a relatively small number of KIF21B molecules can restrict mirotubule length and promote an imbalance of dynein-mediated pulling forces that allows the centrosome to translocate past the nucleus. We conclude that proper control of microtubule length is important for allowing rapid remodeling of the cytoskeleton and efficient T cell polarization.

**\*For correspondence:**
f.m.berger@uu.nl (FB);
a.akhmanova@uu.nl (AA)

[†]These authors contributed equally to this work

**Present address:** [‡]Netherlands Translational Research Center B. V., Oss, Netherlands; [§]Laboratory of Neurovascular Signaling, Department of Molecular Biology, Universite´ libre de Bruxelles (ULB), Gosselies, Belgium

## Introduction

Large-scale reorganization of the microtubule (MT) cytoskeleton is essential for a variety of processes, such as cell division, differentiation and polarization. In dividing cells, a balance of dynein-dependent pulling forces and MT pushing forces determines the positioning of the MT-based mitotic spindle (*Howard and Garzon-Coral, 2017*; *Kotak and Gönczy, 2013*). In interphase cells with dense, non-centrosomal MT arrays, such as epithelial or neuronal cells, polarization is a slow process (reviewed in *Kapitein and Hoogenraad, 2015*; *Meiring et al., 2020*). In contrast, interphase cells with sparse centrosome-based networks can rapidly switch polarity and reposition their MTs. For example, when a T cell encounters an antigen-presenting cell (APC), it translocates its centrosome and associated MTs toward the APC-contact side (*Geiger et al., 1982*; *Kupfer et al., 1983*; *Stinchcombe et al., 2006*; *Yi et al., 2013*) and forms an immunological synapse (reviewed in *Dustin et al., 2010*), a highly specialized compartment encircled by an actin-rich ring. The immunological synapse facilitates T-cell signaling and polarized secretion of cytokines or lytic molecules toward the APC (*Figure 1A*). The process of centrosome repositioning takes only a few minutes and is driven by membrane-associated dynein that pulls MTs with the attached centrosome toward the synapse (*Combs et al., 2006*; *Liu et al., 2013a*; *Martin-Cófreces et al., 2008*; *Nath et al., 2016*).

Although the role of dynein as force generator in centrosome translocation in T cells is firmly established, little is known about the properties of the MT network and MT dynamics that facilitate this rapid cytoskeletal reorganization, and only a few MT regulators participating in this process

**eLife digest** The immune system is composed of many types of cells that can recognize foreign molecules and pathogens so they can eliminate them. When cells in the body become infected with a pathogen, they can process the pathogen's proteins and present them on their own surface. Specialized immune cells can then recognize infected cells and interact with them, forming an 'immunological synapse'. These synapses play an important role in immune response: they activate the immune system and allow it to kill harmful cells.

To form an immunological synapse, an immune cell must reorganize its internal contents, including an aster-shaped scaffold made of tiny protein tubes called microtubules. The center of this scaffold moves towards the immunological synapse as it forms. This re-orientation of the microtubules towards the immunological synapse is known as 'polarization' and it happens very rapidly, but it is not yet clear how it works.

One molecule involved in the polarization process is called KIF21B, a protein that can walk along microtubules, building up at the ends and affecting their growth. Whether KIF21B makes microtubules grow more quickly, or more slowly, is a matter of debate, and the impact microtubule length has on immunological synapse formation is unknown.

Here, Hooikaas, Damstra et al. deleted the gene for KIF21B from human immune cells called T cells to find out how it affected their ability to form an immunological synapse. Without KIF21B, the T cells grew microtubules that were longer than normal, and had trouble forming immunological synapses. When the T cells were treated with a drug that stops microtubule growth, their ability to form immunological synapses was restored, suggesting a role for KIF21B. To explore this further, Hooikaas, Damstra et al. replaced the missing KIF21B gene with a gene that coded for a version of the protein that could be seen using microscopy. This revealed that, when KIF21B reaches the ends of microtubules, it stops their growth and triggers their disassembly. Computational modelling showed that cells find it hard to reorient their microtubule scaffolding when the individual tubes are too long. It only takes a small number of KIF21B molecules to shorten the microtubules enough to allow the center of the scaffold to move.

Research has linked the KIF21B gene to autoimmune conditions like multiple sclerosis. Microtubules also play an important role in cell division, a critical process driving all types of cancer. Drugs that affect microtubule growth are already available, and a deeper understanding of KIF21B and microtubule regulation in immune cells could help to improve treatments in the future.

have been identified. Among them, MT-associated protein MAP4 was shown to be required for centrosome translocation (*Bustos-Morán et al., 2017*), potentially due to its ability to regulate dynein (*Samora et al., 2011*; *Semenova et al., 2014*; *Seo et al., 2016*). In addition, both acetylation and detyrosination of MTs and a complex of casein kinase Iδ with EB1 were suggested to be important for efficient centrosome polarization (*Andrés-Delgado et al., 2012*; *Serrador et al., 2004*; *Zyss et al., 2011*). Finally, a study describing the knockout of stathmin/OP18, a MT-destabilizing protein, in mouse cytotoxic T cells showed decreased centrosome polarization efficiency (*Filbert et al., 2012*). Yet, how these cytoskeletal perturbations affect the MT network of T cells as a whole and how this would consequently hinder centrosome translocation is poorly understood.

An interesting candidate to regulate MT dynamics in immune cells is kinesin-4 KIF21B. Kinesin-4 family members are well known for their ability to limit MT growth in a variety of cellular processes (*Bieling et al., 2010*; *Bringmann et al., 2004*; *He et al., 2014*; *Yue et al., 2018*). Among them are two closely related motors KIF21A and KIF21B, which consist of an N-terminal motor domain, several coiled-coil regions and a C-terminal WD40 domain (*Marszalek et al., 1999*). KIF21A is expressed in many tissues; it is well studied because point mutations impairing its auto-inhibition lead to congenital fibrosis of the extraocular muscles type 1 (CFEOM1), a neurodevelopmental disorder that affects eye movement (*Bianchi et al., 2016*; *Cheng et al., 2014*; *Traboulsi and Engle, 2004*; *van der Vaart et al., 2013*; *Yamada et al., 2003*). In vitro, KIF21A slows down MT growth and suppresses catastrophes, whereas in cells it prevents overgrowth of MTs at the cell cortex (*van der Vaart et al., 2013*).

KIF21B is expressed in brain, eyes, and spleen (*Marszalek et al., 1999*). Several studies have linked the *KIF21B* gene to neurodevelopmental and immune-related disorders, including multiple

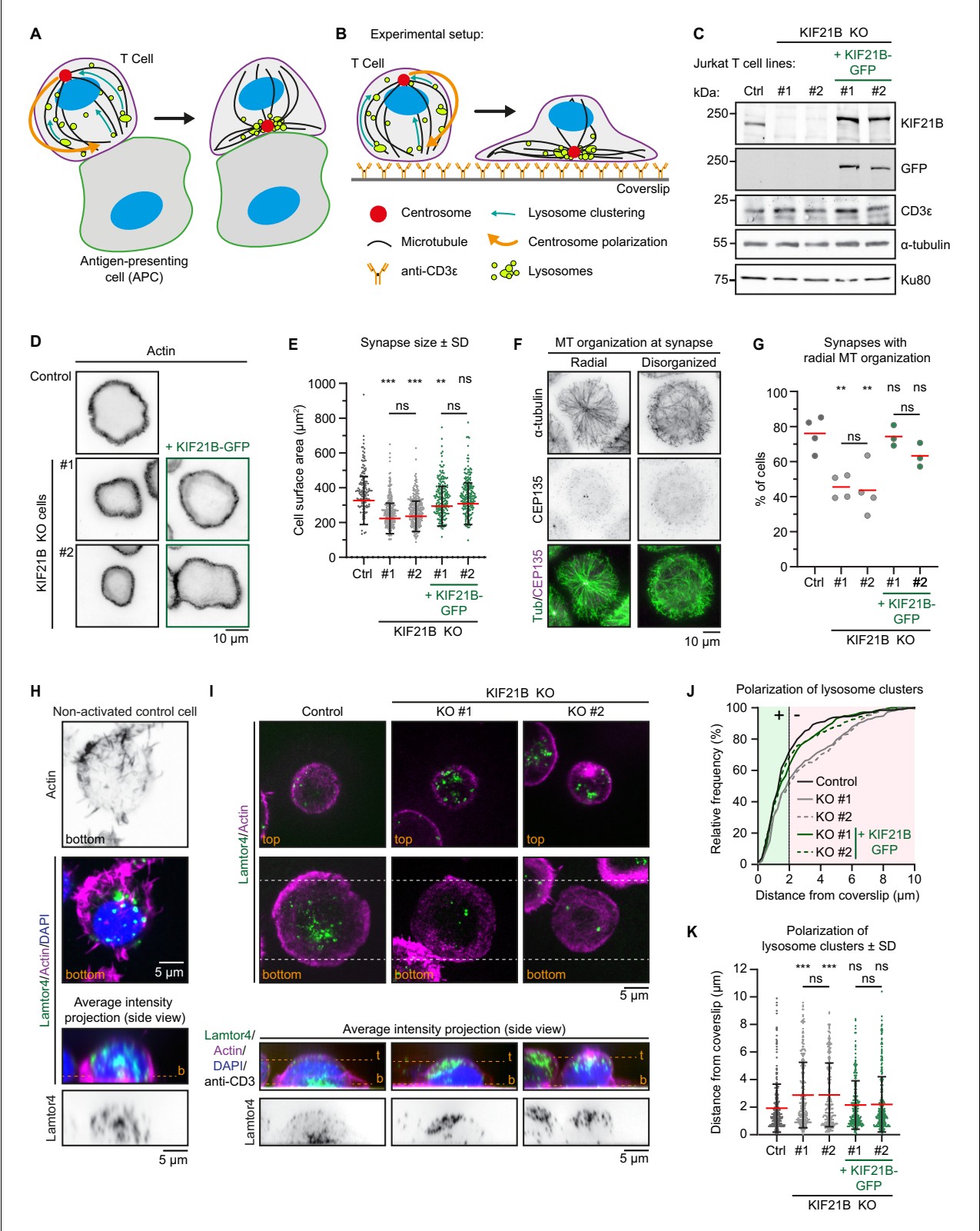

**Figure 1.** Immunological synapse formation is impaired in KIF21B-KO Jurkat T cells. (A–B) Schematic representation of T cells forming an immunological synapse upon target recognition. In vivo, T cells recognize an antigen-presenting cell (APC) via the T cell receptor (CD3)-complex which triggers T cell activation. Upon activation, several organelles including lysosomes are transported to the centrosome by dynein. Simultaneously, the centrosome and its associated organelle cluster polarize to the synapse through MT pulling by membrane-anchored dynein at the synapse. (B) To

*Figure 1 continued on next page*

*Figure 1 continued*

achieve spatiotemporal control on T cell activation, we used anti-CD3-coated glass surfaces to mimic the APC and induce T cell activation and centrosome polarization. (C) Western blot analysis of the indicated Jurkat knockout (KO) cell lines with indicated antibodies. KIF21B-KO clone #1 and #2 were transduced with a KIF21B-GFP construct to generate two polyclonal cell lines re-expressing full-length KIF21B. (D) Phalloidin staining to show F-actin structures in indicated Jurkat T cell lines. Cells were added to poly-D-lysine-coated coverslips with immobilized anti-CD3, fixed after 10 min incubation and imaged on a widefield microscope. (E) Quantification of synapse size expressed as surface area based on a F-actin staining of indicated Jurkat T cells. n = 134, 218, 241, 185 and 194 cells from three independent experiments. **p=0.0098, ***p<0.001, ns = not significant (Mann-Whitney U test). (F) Immunostaining of Jurkat T cells showing two examples of MT organization at the immunological synapse. Cells were added to poly-D-lysine-coated coverslips with immobilized anti-CD3 and fixed 10 min after incubation. Cells were stained for α-tubulin and CEP135 and were imaged on a TIRF microscope. (G) Quantification of MT organization at immunological synapses. Cells were scored per condition either as 'radial' or 'disorganized' depending their MT organization as shown in E. n = 4, 4, 4, 3 and 3 experiments. In total, 373, 389, 306, 367, and 354 cells were quantified per condition. From left to right: **p=0.0023, **p=0.0099, ns = not significant (t test). (H) Confocal images of a non-activated Jurkat T cell on a poly-D-lysine-coated coverslip (without anti-CD3) stained for Lamtor4, F-actin and DAPI. Single Z-plane images (upper panels) show the bottom part of the cell. The position of the single Z-plane images is indicated in the average intensity z-projection (bottom panels) with a dashed line (orange) labeled with 'b' (bottom z-plane). (I) Confocal images of indicated Jurkat T cells. Cells were added to poly-D-lysine-coated coverslips with immobilized anti-CD3 and fixed 10 min after incubation. Cells were stained for Lamtor4, F-actin, and DAPI. In addition, the anti-CD3 antibody with which the coverslip was coated was also stained with a Alexa-dye-conjugated secondary antibody. Single Z-plane images (upper panels) show the top part of the cell and the bottom synapse part (indicated in orange). Dashed lines (white) indicated the area of the average intensity Z-projections shown in the bottom panels. The position of the single Z-plane images shown in the upper panels is indicated with dashed lines (orange) labeled with 't' (top z-plane) and 'b' (bottom z-plane, synapse). (J–K) Quantification of lysosome positioning relative to the synapse-coverslip interface in indicated conditions. The z-positions of lysosome clusters were analyzed by identifying the peak Lamtor4 fluorescence intensity per cell analyzed from an average intensity side-projection of the cell. Cumulative frequency distribution (J) shows the percentage of cells with fully polarized Lamtor4 clusters. A 2 μm threshold was used to distinguish successful (green area) from failed (red area) Lamtor4 cluster polarization events. Dot plot (K) shows all individual cells quantified in (J). n = 318, 284, 244, 216, and 281 cells from four independent experiments. ***p<0.001, ns = not significant (Mann-Whitney U test).

The online version of this article includes the following source data and figure supplement(s) for figure 1:

**Source data 1.** An Excel sheet with numerical data on the quantification of indicated Jurkat cell lines analyzed for cell surface area, MT organization at the immunological synapse, and polarization of lysosomes toward the immunological synapse represented as plots (or cumulative frequency distribution for *Figure 1J*) in *Figure 1E,G,J and K*.

**Figure supplement 1.** Scans of Western blot images corresponding to *Figure 1C*.

**Figure supplement 2.** Characterization of KIF21B knockout cell lines.

**Figure supplement 2—source data 1.** An Excel sheet with numerical data on the quantification of indicated Jurkat cell lines analyzed for IL-2 expression levels after activation, total cell volume, and duration of immunological synapse formation represented as plots in *Figure 1—figure supplement 2A, B, D and G*.

sclerosis and inflammatory conditions such as Crohn's disease and ankylosing spondylitis (*Anderson et al., 2009*; *Asselin et al., 2020*; *Barrett et al., 2008*; *Garcia-Etxebarria et al., 2016*; *Goris et al., 2010*; *International Multiple Sclerosis Genetics Consortium (IMSGC), 2010*; *Kannan et al., 2017*; *Kreft et al., 2014*; *Li et al., 2017*; *Liu et al., 2013b*; *Robinson et al., 2015*; *Yang et al., 2015*). *Kif21b* knock out mice are viable but show behavioral deficits and defects in synaptic transmission (*Ghiretti et al., 2016*; *Gromova et al., 2018*; *Morikawa et al., 2018*; *Muhia et al., 2016*). At the cellular level, KIF21B has been reported to be a processive kinesin involved in neuronal transport (*Ghiretti et al., 2016*; *Gromova et al., 2018*; *Labonté et al., 2014*) and Rac1 inactivation (*Morikawa et al., 2018*). Furthermore, similar to all other kinesin-4 family members, KIF21B acts as a MT growth regulator both in vitro and in cells (*Ghiretti et al., 2016*; *Muhia et al., 2016*; *van Riel et al., 2017*). However, the activity of this kinesin is a matter of controversy, because one study reported that it acts as a positive regulator of MT dynamicity by increasing MT growth rate and catastrophe frequency; in these assays, KIF21B was present at relatively high concentrations (50–300 nM) and was mainly observed to bind to depolymerizing MT ends and lattices (*Ghiretti et al., 2016*). In contrast, our own work showed that KIF21B walks to MT plus ends where it slows down MT growth and potently induces MT pausing already at low nanomolar concentrations, with only one or two KIF21B molecules being sufficient to block elongation of a MT plus end (*van Riel et al., 2017*). Somewhat conflicting results have also been reported on the effects of the loss of KIF21B on MT plus-end growth in neurons: two different studies found opposite effects on MT growth rates, although they agreed that KIF21B reduces MT growth processivity (*Ghiretti et al., 2016*; *Muhia et al., 2016*). Therefore, further analyses of KIF21B behavior and function are required to elucidate its role in regulating MT dynamics.

Here, we have investigated the role of KIF21B in regulating MT organization and dynamics in T cells. We generated KIF21B knockout (KIF21B-KO) Jurkat T cell lines, which displayed defects during immunological synapse formation, particularly in translocating their centrosome toward the synapse. These defects were rescued by KIF21B-GFP re-expression, and live-cell imaging of single KIF21B-GFP motors showed that they walked to MT plus ends and often induced MT pausing followed by depolymerization, very similar to our previous in vitro observations (*van Riel et al., 2017*). Using expansion microscopy, we visualized three-dimensional MT organization in Jurkat T cells and found that MTs were longer in KIF21B-KO cells. Interestingly, we could rescue centrosome polarization defects in KIF21B-KO cells by mildly increasing MT catastrophe frequency with a low dose of vinblastine, a MT-depolymerizing drug. These results were recapitulated in a two-dimensional computational model demonstrating that long MTs impaired centrosome polarization by altering the balance of dynein-driven pulling forcing in a way that prevents symmetry breaking, which could be restored by introducing more frequent catastrophes that limit MT length. We conclude that proper control of MT growth plays a critical role in T cell polarization.

## Results

### KIF21B knockout T cells are defective in centrosome polarization after activation

To study MT reorganization during immune synapse formation, we used Jurkat T lymphocyte cells. We mimicked in vivo T-cell activation by exposing these cells to anti-CD3 coated surfaces, as described previously (*Bunnell et al., 2001*; *Parsey and Lewis, 1993*; *Figure 1A,B*). We confirmed KIF21B protein expression in these cells by Western blotting (*Figure 1C*, *Figure 1—figure supplement 1*). Using CRISPR/Cas9 technology we generated KIF21B knockout cells and selected two clones (#1 and #2) that were analyzed further. To perform rescue experiments, these two knockout cell lines were infected with a lentivirus expressing full-length KIF21B-GFP and the obtained polyclonal cell lines were sorted for GFP-positive cells and analyzed by Western blotting (*Figure 1C*, *Figure 1—figure supplement 1*). KIF21B expression was lost in both knockout lines, whereas KIF21B-GFP expression mildly exceeded endogenous KIF21B. Expression of α-tubulin was not affected in these lines. In addition, we established that these Jurkat T cell lines were responsive to stimulus by first confirming the presence of CD3ε, a subunit of the T cell receptor (TCR) (*Figure 1C*, *Figure 1—figure supplement 1*). Next, we stimulated control and KIF21B-KO cells with phorbol 12-myristate 13-acetate (PMA) and ionomycin, which increase the downstream intermediates of TCR signaling, protein kinase C activity, and intracellular calcium levels, respectively. This stimulation regime upregulates Interleukin-2 (IL-2) mRNA expression, a hallmark of T cell activation, and we found that our knockout cell lines responded similarly to control cells (*Figure 1—figure supplement 2A,B*).

To monitor immunological synapse formation, we added Jurkat cells to anti-CD3 coated coverslips, fixed the cells 10 min later and stained them with phalloidin to visualize the peripheral actin ring (*Figure 1B,D*). We measured synapse size and noticed that KIF21B-KO cell lines formed 28.4% (KO #1) and 27.3% (KO #2) smaller synapses compared to control cells and that this phenotype could be partially rescued by re-expression of KIF21B-GFP (*Figure 1E*). The reduction in the size of immunological synapses in KIF21B-KO cells was not due to smaller overall size of these cells, as their volume was the same as that of control cells (*Figure 1—figure supplement 2C,D*). Live-cell imaging using Differential Interference Contrast (DIC) microscopy showed that synapse formation was slower in KIF21B-KO cells (*Figure 1—figure supplement 2E–G*).

Next, we analyzed the overall MT organization in these cells by Total Internal Reflection Fluorescence (TIRF) microscopy, which visualizes a thin optical section located above the coverslip. In fully polarized cells, the centrosome-centered MT aster was visible in the TIRF focal plane, whereas in cells that failed to polarize properly, the centrosome was not visible by TIRF, and MTs located in the vicinity of the coverslip appeared disorganized (*Figure 1F*). In this experiment, the centrosome was located in the plane of the synapse in the majority of control (76%) and KIF21B-GFP-expressing cells (74.3% and 63.3%), whereas this was the case only in 45.5% (KO #1) and 43.6% (KO #2) of KIF21B-KO cells (*Figure 1G*).

During T cell activation, lysosomes cluster around the centrosome and relocate together with it toward the synapse (*Figure 1A,B*). Immunostaining of non-activated Jurkat cells on poly-D-lysine-

coated coverslips showed that lysosomes were scattered throughout the cytoplasm. We note that although it has been reported that cationic surfaces are sufficient to initiate T cell signaling (*Santos et al., 2018*), in our hands, exposure to poly-D-lysine-coated coverslips did not trigger centrosome polarization or lysosome clustering (*Figure 1H*, *Figure 1—figure supplement 2H,I*). In contrast, after activation on anti-CD3-coated coverslips, both control and knockout cells showed lysosome clustering, indicating that KIF21B knockout does not affect the minus-end-directed transport of lysosomes (*Figure 1H,I*). We then analyzed the position of the lysosome cluster relative to the coverslip using z-projections of these cells and found that only 52.4% KIF21B-KO #1% and 50.4% KIF21B-KO #2 cells contained a polarized lysosome cluster compared to 71% of control cells (*Figure 1I,J*). Again, we could rescue this phenotype by re-expressing KIF21B-GFP (*Figure 1J,K*). These data suggest that KIF21B-KO Jurkat cells exhibit defects in immunological synapse formation: they are less efficient in polarizing their centrosome and the associated lysosomes.

## MT organization is altered in KIF21B knockout T cells

We next examined the organization of the MT network of control and KIF21B-KO T cells in more detail using Stimulated Emission Depletion (STED) microscopy. For optimal labeling of MTs, cells were added to anti-CD3-coated coverslips for 7 min followed by pre-permeabilization, cytoplasm extraction, and fixation as described before (*Tas et al., 2018*). To compare MT organization in the presence or absence of KIF21B, we focused on fully polarized cells, despite centrosome polarization often being impaired in KIF21B-KO cells (*Figure 1F,G*). We found that while in control cells most MTs were organized in an aster-like pattern, the knockout cells often contained circular MT bundles at the cell periphery. To analyze this phenotype, we made use of a custom-developed ImageJ plugin (*Martin et al., 2018*) to separate our images into radial and non-radial components based on MT orientation in relation to the centrosome (*Figure 2B*). This analysis showed that KIF21B-KO cells have an increased proportion of non-radially oriented MTs, which are organized into peripheral MT bundles at the immunological synapse (*Figure 2A–C*).

To analyze whether the MT network as a whole is affected by KIF21B depletion, we used Expansion Microscopy (ExM) to visualize the complete MT network and distinguish single MT polymers in 3D. With ExM, samples are physically magnified by embedding fixed cells in a isotropically swellable polymer, giving an approximate 4.5x-fold increase in resolution in x, y, and z (*Chen et al., 2015*; *Jurriens et al., 2020*). To understand how the state of the MT network before T cell activation may affect centrosome translocation, we looked at the very early stage of immunological synapse formation. T cells were fixed after 2 min of activation, when the majority of control cells did not yet fully polarize. We noticed that the MT network appeared to be denser in KIF21B-KO cells compared to control cells (*Figure 2D,E*, *Figure 2—videos 1–3*). To exclude that this was caused by increased levels of MT nucleation, we counted the number of MTs originating from the centrosome by analyzing the number of filaments passing through a 1-μm-thick spherical shell with an inner radius of 5 μm generated around the centrosome (*Figure 2D*, *Figure 2—videos 1–3*). We found that MT number was slightly reduced in KIF21B-KO cells, by 21.7% in KO #1% and 19% in KO #2 (*Figure 2F*). Since T cells have highly centrosome-focused MT arrays, we assumed that all MTs are anchored to the centrosome and that these numbers thus represent the total number of MTs per cell. Next, we estimated an average MT length per cell. We generated surface masks based on the colocalization of the MTs and the 1-μm-thick spherical shell around the centrosome (*Figure 2D*, red surfaces). Since MTs cross through the shell orthogonally to its surface, the mean intensity corresponding to 1 μm MT was determined by averaging the fluorescence intensity per surface adjusted for the number of MTs in each surface. We then used a surface mask to determine the total fluorescence intensity corresponding to the complete MT network and divided it by the MT intensity per micron and by the number of MTs. We found that KIF21B-KO cells contain fewer MTs but they are on average longer than those in control cells (*Figure 2E–G*). Since the overall MT mass is higher in KIF21B-KO cells (*Figure 2—figure supplement 1A,B*), the concentration of free tubulin in these cells might be lower; this would affect MT nucleation at the centrosome, causing a reduction in MT number.

We furthermore examined the geometry of MTs in polarizing control and KIF21B-KO cells (*Figure 2—figure supplement 1B*). Considering the immunological synapse as the basal side of the cell, with nucleus located above it, the centrosome could be positioned on top or on the side of the nucleus, or between the nucleus and the synapse. In agreement with the previously described observations (*Yi et al., 2013*), a MT bundle directed from the centrosome to the synapse ('stalk') could be

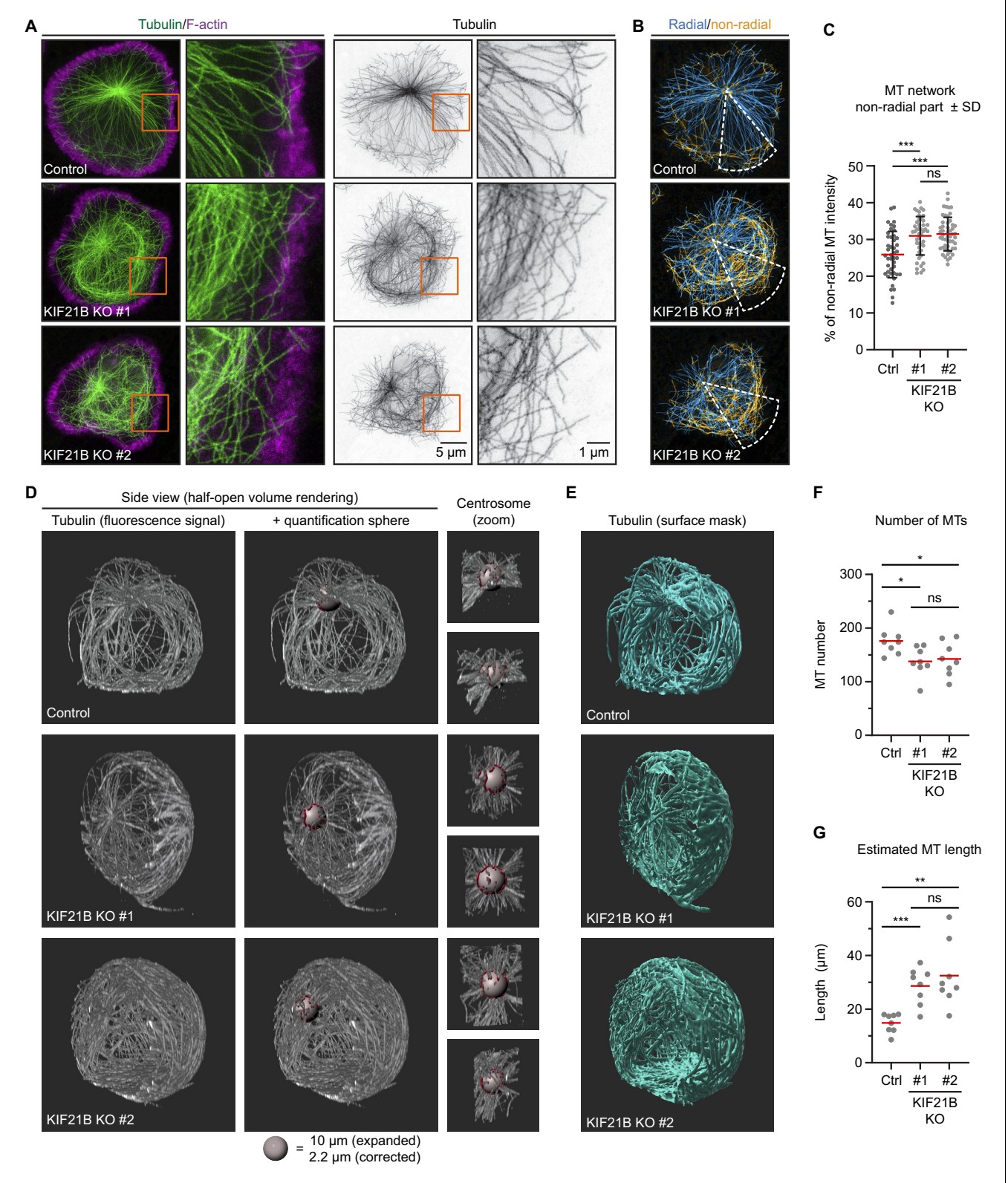

**Figure 2.** Knockout of KIF21B causes microtubule (MT) overgrowth in Jurkat cells. (A–B) STED images of indicated Jurkat T cell lines. Cells were added to poly-D-lysine-coated coverslips with immobilized anti-CD3 and fixed 7 min after incubation. Cells were stained for α-tubulin and F-actin. Orange boxes indicate zoomed areas shown on the right. (B) Images were background subtracted and split into radial and non-radial components based on the MT orientation in relation to the centrosome. Radial intensity profiles were made from a 45° sector starting from the centrosome and directed
*Figure 2 continued on next page*

*Figure 2 continued*

toward the farthest removed portion of the cell periphery (white). The resulting heat maps were used to quantify the proportion of non-radially oriented MTs as described in Materials and methods. (C) Quantification of the proportion of non-radially oriented MTs analyzed per cell. n = 44, 45, and 52 cells from three independent experiments. ***p<0.001, ns = not significant (t test). (D–E) Volume renderings of indicated T cell lines stained for α-tubulin. T cells were fixed 2 min after activation on poly-D-lysine-coated coverslips with immobilized anti-CD3. Samples were expanded ~4.5 times following an Expansion Microscopy (ExM) protocol and MTs were imaged using on a confocal microscope. Imaris software was used to create volume renders and analysis. MTs emanating from the centrosome were analyzed using a spherical shell with a 10 µm inner diameter and a thickness of 1 µm (gray); filaments crossing the sphere are highlighted (red). Zooms of the centrosome are shown from two different angles. (E) Fluorescence signal was converted to a surface mask using Imaris software to analyze total tubulin intensity. Additional examples are shown in *Figure 2—figure supplement 1B*. (F) Quantification of the total number of MTs per cell emanating from the centrosome quantified at 5 µm distance from the centrosome of indicated Jurkat T cell lines. n = 8 cells for all conditions. From left to right: *p=0.0131, *p=0.0367, ns = not significant (t test). (G) Quantification of estimated MT length per cell in indicated Jurkat T cell lines. Fluorescence intensity of 1 µm filament length was measured for all MTs crossing a spherical shell of 1 um width with a 5 µm inner radius located around the centrosome, as shown in D. The average MT length per cell was calculated by correcting the total fluorescence intensity per cell by the number of MTs at the centrosome. n = 8 cells for all conditions. ***p=0.0002, **p=0.0013, ns = not significant (t test).

The online version of this article includes the following video, source data, and figure supplement(s) for figure 2:

**Source data 1.** An Excel sheet with numerical data on the quantification of indicated Jurkat cell lines analyzed for MT radiality at the immunological synapse and total 3D MT organization of Jurkat cells during polarization represented as plots in *Figure 2C,F and G*.

**Figure supplement 1.** Characterization of microtubule (MT) network in polarizing T cells by Expansion Microscopy (ExM).

**Figure supplement 1—source data 1.** An Excel sheet with numerical data on the quantification of indicated Jurkat cell lines analyzed for total MT length in 3D of Jurkat cells during polarization represented as a plot in *Figure 2—figure supplement 1A*.

**Figure 2—video 1.** 3D volume render of an Expansion Microscopy (ExM) imaged Jurkat control cell 2 min after activation.
https://elifesciences.org/articles/62876#fig2video1

**Figure 2—video 2.** 3D volume render of an Expansion Microscopy (ExM) imaged Jurkat KIF21B-KO cell (KO #1) 2 min after activation.
https://elifesciences.org/articles/62876#fig2video2

**Figure 2—video 3.** 3D volume render of an Expansion Microscopy (ExM) imaged Jurkat KIF21B-KO cell (KO #2) 2 min after activation.
https://elifesciences.org/articles/62876#fig2video3

found in some cells (*Figure 2—figure supplement 1C*). We note that it was not obvious from our images that all MTs in such a stalk terminated at synapse; rather it appeared that many of them made side contacts with the membrane. In other cases, a single basally directed MT bundle could not be distinguished (*Figure 2—figure supplement 1B*), possibly because these cells were at an earlier stage of centrosome polarization. In cells with apically located centrosomes, MTs appeared to embrace the nucleus from all sides and then make lateral contacts with the synapse surface. In KIF21B-KO cells, such MTs formed a dense network in the basal half of the cell (*Figure 2—figure supplement 1B*).

To summarize, T cells that lack KIF21B show MT overgrowth at the immunological synapse. Using 3D ExM, we confirmed this observation and showed that KIF21B-KO cells have overly long MTs that form dense arrays at the immunological synapse.

## KIF21B is a processive motor that induces MT pausing and catastrophes

Next, we investigated the localization and dynamics of KIF21B by live-cell imaging of the KIF21B-KO cell line rescued by reexpressing KIF21B-GFP. In HeLa cells, the closely related paralogue KIF21A displays strong peripheral localization at cortical MT stabilizing complexes (CMSC), to which it binds through KANK proteins (*Noordstra and Akhmanova, 2017*; *van der Vaart et al., 2013*). Structural studies have elucidated the binding interface of this interaction, and it is clear that KIF21B does not possess the KANK-binding region required for association with the CMSC (*Guo et al., 2018*; *Pan et al., 2018*; *Weng et al., 2018*). Likewise, KIF21B-GFP did not show any obvious cortical accumulation in T cells but was rather diffusely distributed in the cytoplasm or associated to MTs on which it moved processively (*Figure 3A*, *Figure 3—video 1*). The average velocity of KIF21B-GFP puncta was 0.77 ± 0.14 µm/s (mean ± SD) (*Figure 3A,B*), which is 1.2 and almost four times faster compared to velocities measured for the full-length motor in COS-7 cells and hippocampal neurons, respectively (*Ghiretti et al., 2016*; *van Riel et al., 2017*).

To investigate whether the observed KIF21B-GFP signals correspond to single kinesin dimers, we compared their fluorescence intensity to that of single GFP molecules, as described previously for in

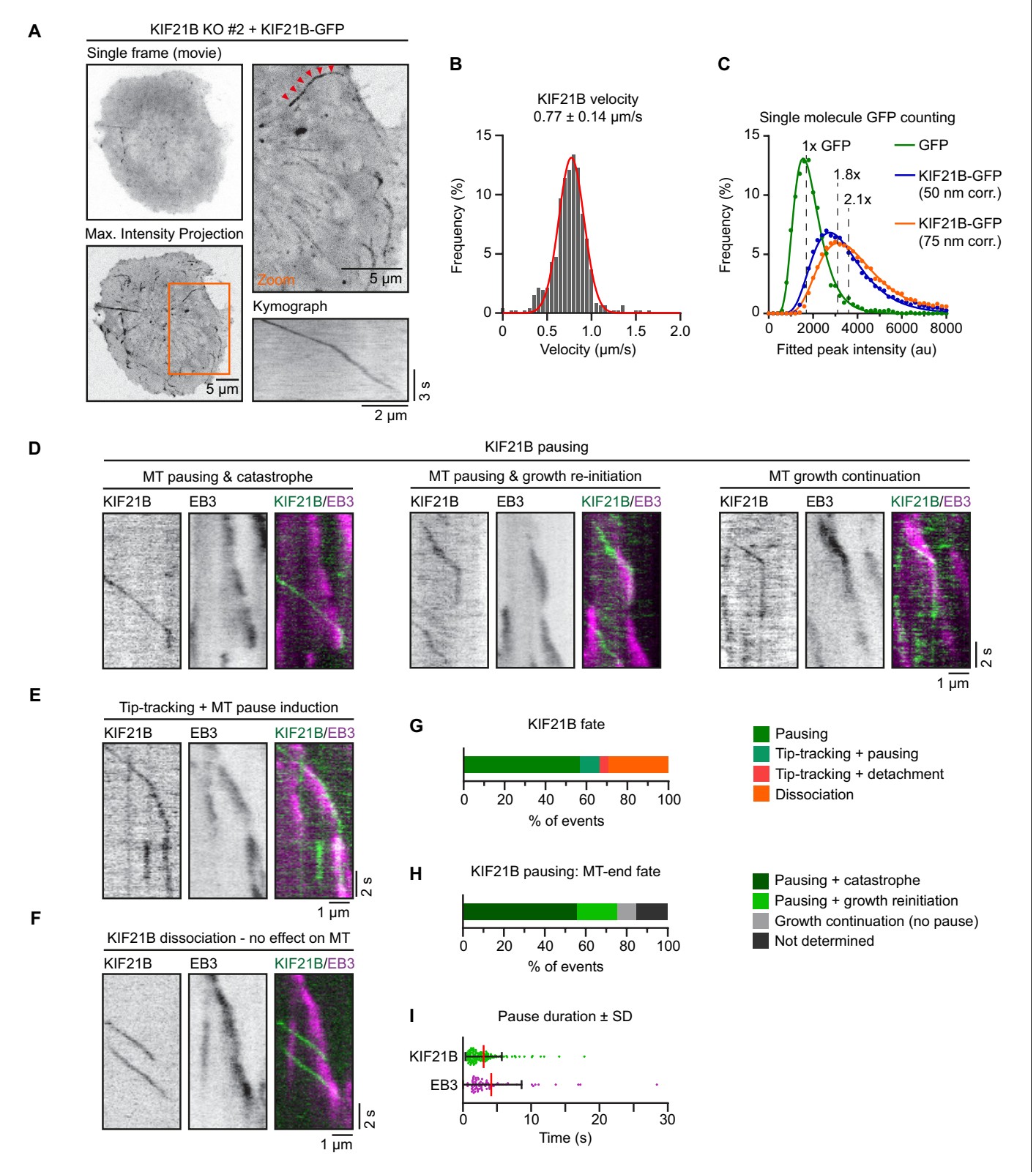

**Figure 3.** KIF21B induces microtubule (MT) pausing and catastrophes in T cells. (**A**) Live imaging of indicated Jurkat KIF21B-KO T cells stably overexpressing KIF21B-GFP on Lab-Tek chambered coverglass with immobilized anti-CD3. Images show a single movie frame and a maximum intensity projection of a background-subtracted movie. Zoomed areas are indicated (orange), and the kinesin track corresponding to the kymograph shown is indicated with red arrowheads. (**B**) Frequency distribution of KIF21B-GFP velocity in cells measured from data obtained imaging of KIF21B-KO #2 Jurkat

*Figure 3 continued on next page*

*Figure 3 continued*

T cell line stably overexpressing KIF21B-GFP on a TIRF microscope at 10 fps. Velocities were measured by kymograph analysis and the numbers were fitted to a Gaussian (red curve). n = 464 events from 102 cells from four independent experiments. (C) Histograms (dots) of peak-fitted fluorescence intensities of monomeric GFP immobilized on glass (green dots) and KIF21B-GFP motors imaged in Jurkat T cells (blue/orange) in two separate chambers on the same coverslip and the corresponding lognormal fits (solid lines). Imaging conditions were identical for both samples. KIF21B-GFP intensity values were corrected for a 50 and 75 nm distance from the coverslip, as illustrated in *Figure 3—figure supplement 1B*. n = 2601 (GFP) and n = 16523 (6 cells) (KIF21B-GFP) molecules. Mean values relative to monomeric GFP are indicated in the plot. (D–I) Live-cell imaging of KIF21B-KO #2 Jurkat T cell line stably overexpressing KIF21B-GFP and transiently overexpressing EB3-mCherry to label growing MT plus ends. Cells were added on Lab-Tek chambered coverglass with immobilized anti-CD3 and imaged on a TIRF microscope at 10 fps. (D) Kymographs illustrating KIF21B-GFP motors switching from walking to a pausing state. Growing MTs are visualized by EB3-mCherry overexpression. Examples show a growing plus end that undergoes pausing after KIF21B-GFP arrival followed by a catastrophe event (left), or growth re-initiation when the pausing KIF21B-GFP dissociates (middle). Some plus ends are not affected by the KIF21B-GFP motor pause event (right). (E) Kymograph illustrating a KIF21B-GFP tip-tracking on a growing EB3-mCherry plus end followed by a pausing event of both KIF21B-GFP and the MT plus end. (F) Kymograph illustrating two KIF21B-GFP motors reaching a growing EB3-mCherry plus-end causing the motors to dissociate from the MT plus-end, which continues growing, though EB3-mCherry signal is reduced. (G) Quantification of KIF21B-GFP fates observed when a moving KIF21B motor reaches an EB3-mCherry-labeled MT plus end, as illustrated in D-F. n = 224 events from two independent experiments. (H) Quantification of MT plus-end fates when a KIF21B-GFP motor reaches a growing EB3-mCherry-labeled MT plus end and transitions to a paused state, as illustrated in D-F, *Figure 3—figure supplement 1C*. n = 149 events from two independent experiments. (I) Quantification of pause duration of KIF21B-GFP and EB3-mCherry-labeled MT plus ends, illustrated in G-H. n = 144 and 75 events from two independent experiments.

The online version of this article includes the following video, source data, and figure supplement(s) for figure 3:

**Source data 1.** An Excel sheet with numerical data on the quantification of KIF21B-GFP velocities, single-molecule analysis of KIF21B-GFP in Jurkat cells and the effects of single KIF21B-GFP molecules on growing microtubule ends in Jurkat cells represented as frequency distributions (*Figure 3B*), frequency distributions with lognormal fits (*Figure 3C*), and as plots (*Figure 3G, H and I*).

**Figure supplement 1.** Characterization of KIF21B motility in cells.

**Figure supplement 1—source data 1.** An Excel sheet with numerical data on the quantification of single-molecule analysis of KIF21B-GFP in Jurkat cells and the effects of single KIF21B-GFP molecules on growing microtubule ends in Jurkat cells represented as a lognormal data fit (*Figure 3—figure supplement 1B*) and as plots (*Figure 3—figure supplement 1B and F*).

**Figure 3—video 1.** KIF21B-GFP motors imaged at the immunological synapse of an activated Jurkat T cell.

https://elifesciences.org/articles/62876#fig3video1

---

vitro reconstitution assays (*Aher et al., 2018*; *Hooikaas et al., 2019*; *van Riel et al., 2017*). We imaged dual chamber slides that contained purified monomeric GFP in one chamber and activated KIF21B-GFP-expressing T cells in the other chamber. Intracellular KIF21B-GFP motors were detected using DoM software (see Materials and methods for details) and the detected signals were filtered for molecules belonging to tracks ≥ 1 s (*Figure 3—figure supplement 1A*). We found that their signal was ~1.5 times brighter than that of monomeric GFP (*Figure 3C*, *Figure 3—figure supplement 1B*). Given that the evanescent field used for excitation decays exponentially and the KIF21B-GFP motors observed were localized intracellularly, one should take into account the distance between the coverslip and fluorescent molecules. Although we could not measure this distance precisely, we assumed that the thickness of the plasma membrane, cortical actin, cytoplasm and the 25 nm width of a MT would result, on average, in at least 50 nm spacing. Given the penetration depth $d$ in these imaged cells was calibrated at 180 nm, we estimated that the corrected the KIF21B-GFP fluorescence signal intensity was ~2 times brighter than that of monomeric GFP (~1.8 times assuming 50 nm distance from the coverslip or ~2.1 times assuming 75 nm distance) (*Figure 3C*, *Figure 3—figure supplement 1B*). Most of the motile KIF21B-GFP puncta in Jurkat cells thus likely corresponded to single kinesin dimers, which did not form any clear accumulations.

Our previous in vitro work showed that single KIF21B molecules walked toward a growing plus end and induced transient pauses ultimately followed by a catastrophe, whereas multiple KIF21B molecules accumulated at a MT plus end induced prolonged pauses (*van Riel et al., 2017*). Consequently, we asked whether KIF21B behaves similarly in T cells to regulate MT dynamics. To observe the interactions of KIF21B-GFP with MT plus ends in Jurkat cells, we transfected them with EB3-mCherry, which labels growing MT ends and shows some MT lattice decoration, and imaged these cells on a TIRF microscope. We traced processive KIF21B-GFP motors and quantified events where the motor reached the tip of an EB3-mCherry comet. Most of KIF21B-GFP molecules (56.7%) transitioned to a paused state upon reaching the plus end; some motors (29.5%) dissociated from the MT and in a few cases KIF21B tracked the MT tip and then either transitioned to a paused state (9.8%)

or dissociated from the MT (4%) (*Figure 3D–G*). It should be noted that even in cases where KIF21B motor rapidly dissociated from the MT tip, we sometimes could observe a reduction in EB3 comet intensity (*Figure 3F*), suggesting a transient MT growth perturbation. A pausing KIF21B-GFP motor typically did affect MT plus-end growth, and the events where a MT first paused and ultimately switched to depolymerization were most frequent (55.7%). In other cases, a MT first paused but then switched back to growth after the paused KIF21B-GFP motor dissociated from the plus end (19.5%). In a few cases (9.4%), a growing plus end kept growing after a KIF21B-GFP motor transitioned to pausing; often, such events coincided with a reduced EB3 comet intensity (*Figure 3D,G, H*). On average, KIF21B-GFP motors were paused for 3.0 ± 2.7 s (mean ± SD) (*Figure 3I*), whereas MT plus-end pause durations measured with EB3-mCherry were somewhat longer, 4.2 ± 4.4 s (mean ± SD) (*Figure 3I*). This can be explained either by photobleaching of the GFP or by participation of additional cellular factors that could stabilize paused MT ends after KIF21B dissociates. In some cases, growing MT plus end appeared to be static; these events likely corresponded to MTs growing against an obstacle, where MT tip remained stationary while the growing MT shaft buckled and formed a loop (*Figure 3—figure supplement 1C,D*). Events where KIF21B-GFP directly dissociated from a MT end upon arrival affected the growing MT plus end much less compared to KIF21B-GFP pausing events; only a few of such events coincided with the initiation of a MT pause or a direct catastrophe event (*Figure 3—figure supplement 1E,F*). All these data represent events where KIF21B-GFP was observed to arrive to the MT tip by walking along the MT shaft. We also occasionally observed events where KIF21B-GFP directly bound to the MT plus end and affected MT growth, but such events were infrequent and were therefore were not quantified (*Figure 3—figure supplement 1G*). We conclude that KIF21B is a fast and processive kinesin in T cells and that its activity at the plus ends limits growth by inducing MT pausing and catastrophes.

## MTs grow faster and more processively in cells lacking KIF21B

Since we directly observed MT growth perturbation by KIF21B-GFP, we next set out to perform a more global analysis of MT plus-end dynamics in Jurkat cells either expressing or lacking this motor. To this end, we used lentiviral transduction to introduce EB3-GFP as a MT plus-end marker into control and KIF21B-KO cell lines. We used these cells for live imaging of growing MT plus ends in cells that were polarized after activation (*Figure 4A*). We found that the MT growth rate was on average increased by 18% (KO #1) and 15% (KO #2) compared to control Jurkat cells (*Figure 4A,B*). These observations were in line with previous work showing that overexpression of KIF21B-GFP in COS-7 cells, which do not have endogenous KIF21B, reduced MT growth rates by ~1.5-fold (*van Riel et al., 2017*). These data are also consistent with the observation of increased MT growth rate after transient KIF21B depletion in neurons (*Ghiretti et al., 2016*), but contradict the analysis of MT growth speed in *Kif21b* knockout neurons (*Muhia et al., 2016*). Furthermore, we observed a decrease in MT catastrophe rate in KIF21B knockout cells (*Figure 4C*), which fits well with previous studies in neurons (*Ghiretti et al., 2016*; *Muhia et al., 2016*).

To further evaluate MT plus-end fate in cells lacking KIF21B, we infected control and KIF21B-KO cell lines with a GFP-β-tubulin construct as a MT marker. These cells were imaged on a TIRF microscope, and we quantified events where the plus end of a radially growing MT reached the cell periphery. In control cells, a small minority (9.8%) of MTs underwent a catastrophe or continued to grow by sliding along the cell cortex (*Figure 4D,E*). However, most MTs paused in this region (44.2%) or continued their growth and this led to MT buckling, because the MT plus end remained stationary, whereas the increasingly long MT shaft was pushed backwards and formed a loop (40.9%) (*Figure 4D,E*). In KIF21B-KO cells, MTs displayed less pausing at the cell periphery, whereas buckling became more frequent (*Figure 4E*). A closer examination of MT pauses revealed that these events in control cells were significantly more long-lived: 8.3 ± 5.6 s (mean ± SD) compared to those in KIF21B-KO cells: 4.1 ± 2.2 s (KO #1, mean ± SD) and 4.3 ± 2.4 s (KO #2, mean ± SD) (*Figure 4F*, *Figure 4—figure supplement 1A*). We note that these pause durations are longer than those observed with EB3-mCherry (*Figure 3I*), because the latter labels growing MTs, and its signal is gradually lost when MT plus ends switch to pausing and lose the GTP cap that EB3 labels (*Roostalu et al., 2020*). Overall, these data indicate that in T cells, KIF21B acts as a factor that inhibits MT growth and induces pausing and catastrophes.

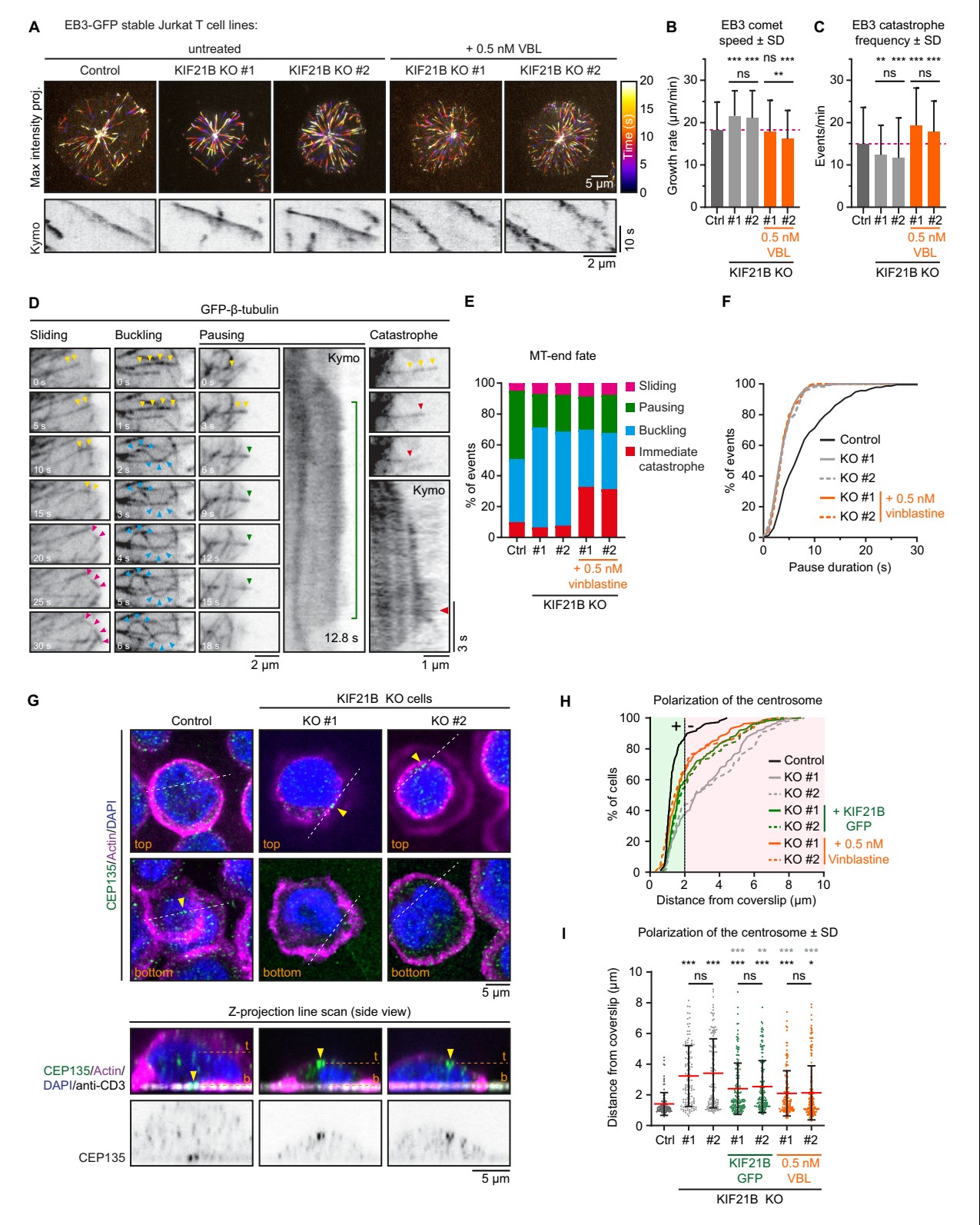

**Figure 4.** Mild inhibition of MT growth with vinblastine rescues centrosome repositioning in KIF21B-KO cells. (**A**) Live-cell imaging of indicated EB3-GFP overexpressing Jurkat T cell lines. Cells were added to Lab-Tek chambered coverglass with immobilized anti-CD3 and imaged on a TIRF microscope at 2.5 fps. Color-coded maximum intensity projections and illustrative kymographs of growing EB3-GFP comets are shown per condition. KIF21B-KO cells were untreated or treated with 0.5 nM vinblastine 30 min prior to imaging as indicated. A scale for the time-dependent color-coding is

*Figure 4 continued on next page*

Figure 4 continued

indicated on the right. (B) Quantification of MT growth rates determined from imaging EB3-GFP in indicated Jurkat T cell lines. Dashed line (magenta) indicates the average growth rate for control cells. n = 470, 478, 250, 255, and 267 events from three independent experiments. **p=0.0024, ***p<0.001, ns = not significant (Mann-Whitney U test). (C) Quantification of MT catastrophe frequencies determined from imaging EB3-GFP in indicated Jurkat T cell lines. Dashed line (magenta) indicates the average catastrophe frequency for control cells. n = 178, 178, 170, 205, and 173 events from two independent experiments. **p=0.0029, ***p<0.001, ns = not significant (Mann-Whitney U test). (D) Live-cell imaging of β-tubulin-GFP overexpressing Jurkat T cell lines. Cells were added on Lab-Tek chambered coverglass with immobilized anti-CD3 and imaged on a TIRF microscope at 10 fps. Movie stills and kymographs show examples of MTs that undergo sliding (pink arrowheads), buckling (cyan arrowheads), pausing (green arrowheads), or catastrophe (red arrowheads) after reaching the cell periphery. Yellow arrowheads indicate growing MTs before the highlighted events take place. Kymographs show the corresponding MT pausing and catastrophe events; the catastrophe initiation (red arrowhead) and total pause time (green line) are indicated. (E) Quantification of MT plus-end fates after reaching the cell periphery based on live-cell imaging of indicated β-tubulin-GFP overexpressing Jurkat T cell lines. n = 518, 508, 477, 544, and 409 events from 46, 49, 38, 37, and 32 cells obtained from three independent experiments. (F) Cumulative frequency distribution shows pause duration of events quantified in E based on live-cell imaging of indicated β-tubulin-GFP overexpressing Jurkat T cell lines. A quantification of these data is shown in *Figure 4—figure supplement 1A*. n = 229, 110, 114, 117, and 101 events from 46, 49, 38, 37, and 32 cells obtained from three independent experiments. (G) Confocal images of indicated Jurkat T cells. Cells were added to poly-D-lysine-coated coverslips with immobilized anti-CD3 and fixed 10 min after incubation. Cells were stained for CEP135, F-actin and DAPI. In addition, the anti-CD3 antibody with which the coverslip was coated was also stained with a Alexa-dye-conjugated secondary antibody. Single Z-plane images (upper panels) show the top part of the cell and the bottom synapse part. White dashed lines indicate the position of the average intensity Z-projections shown in the bottom panels. Yellow arrowheads indicate the position of the CEP135-positive centriole staining. The positions of the single Z-plane images shown in the upper panels are indicated with dashed lines (orange) labeled with 't' (top z-plane) and 'b' (bottom z-plane, synapse). (H–I) Quantification of CEP135 distance from the synapse-coverslip interface for indicated conditions. Cumulative frequency distribution (H) shows the percentage of cells with fully polarized centrosomes. A 2 μm threshold was used to distinguish successful (green area) from failed (red area) centrosome polarization events. Dot plot (I) shows the distance values for individual cells quantified in (H). n = 109, 119, 132, 150, 150, 133, and 175 cells from three independent experiments. Comparison with control (black stars): *p=0.0187, ***p<0.001, ns = not significant (Mann-Whitney U test). Comparison with related KIF21B KO condition (gray stars): **p=0.0042, ***p<0.001, ns = not significant (Mann-Whitney U test).

The online version of this article includes the following source data and figure supplement(s) for figure 4:

**Source data 1.** An Excel sheet with numerical data on the quantification of indicated Jurkat cell lines for MT dynamics parameters and centrosome polarization toward the immunological synapse represented as plots (*Figure 4B, C, E and I*) and cumulative frequency distributions (*Figure 4F and H*).
**Figure supplement 1.** KIF21B-depleted cells lack long MT pause events.
**Figure supplement 1—source data 1.** An Excel sheet with numerical data on the quantification of indicated Jurkat cell lines for MT pause durations represented as a plot in *Figure 4—figure supplement 1A*.

## MT growth inhibition with vinblastine rescues the defects in KIF21B knockout cells

To establish a causal connection between MT overgrowth and defects in immune synapse formation, we sought for a way to mildly inhibit MT growth in KIF21B-KO cells. Our previous work has shown that a low dose of vinblastine mildly perturbs growth and promotes catastrophes but does not induce MT depolymerization (*Bouchet et al., 2016*; *Mohan et al., 2013*). Indeed, at a low (0.5 nM) vinblastine concentration, MT network was preserved and was similar to that in control cells but different from KIF21B-KO cells, as no circular MT bundles were present at the cell periphery (compare *Figure 4—figure supplement 1B* to *Figure 2A*). MT organization in KIF21B-KO cells treated with 0.5 nM vinblastine was similar to that of KIF21B-KO cells rescued with KIF21B-GFP (*Figure 4—figure supplement 1B*). In vinblastine-treated KIF21B-KO cells, MT growth events visualized with EB3-GFP were less continuous, and the growth rates were reduced approximately to the level seen in control Jurkat cells (*Figure 4A–C*). We also examined the effect of 0.5 nM vinblastine on the dynamics of MTs labeled with GFP-β-tubulin and found that vinblastine application resulted in an increased catastrophe frequency, while long-lived MT pauses were not restored (*Figure 4E,F*, *Figure 4—figure supplement 1A*).

We next tested whether such a mild vinblastine-induced MT growth inhibition would rescue the ability of the KIF21B-KO cells to polarize their centrosome. Jurkat cells were added to anti-CD3-coated coverslips and stained for CEP135, a centriole marker, to analyze polarization using z-projections (*Figure 4G*). We scored polarization events as successful when the centrosome was located within 2 μm from the coverslip and found that most control cells (87.2%) were polarized, contrary to knockout cells (36.7% and 42.9%) (*Figure 4H*). This phenotype could be partially rescued by reexpressing KIF21B-GFP in these cells (58.9% and 51.7%), and an even slightly better rescue was observed after applying 0.5 nM vinblastine to both KIF21B-KO cell lines (65.4% and 66.9%)

(*Figure 4H,I*). Altogether, we found that KIF21B depletion alters MT growth dynamics and that centrosome polarization defects in KIF21B-depleted cells can be rescued by the application of a drug, vinblastine, that inhibits MT growth. This indicates that cell polarization defects caused by the absence of KIF21B are due to its effect on MT dynamics rather than potential perturbation of transport or signaling processes.

## Biophysical simulation of MT dynamics suggests that a small number of KIF21B molecules can limit MT overgrowth

To gain more insight into the regulatory effects of KIF21B and to explore possible mechanisms explaining how overly long MTs hinder centrosome polarization, we constructed a biophysical model based on *Cytosim* (*Nedelec and Foethke, 2007*). In contrast to previous more coarse-grained computational studies (*Kim and Maly, 2009*) and less modular models (*Hornak and Rieger, 2020*), we chose *Cytosim* to explicitly investigate the effects of KIF21B on MT dynamics. *Cytosim* uses overdamped Langevin equations to simulate the cytoskeleton with associated proteins and organelles restricted in a defined space. To model a T cell, we assumed a rotationally invariant space and simplified it to a two-dimensional object. Our model cell contains a nucleus and a centrosome, which is confined to one side of the cell and has 90 MTs attached. The cell boundary and the nucleus acted as elastic objects: when deflected, they exert a restoring force. The centrosome was modeled as a rigid sphere with uniformly spread MT anchoring points with a rotational elasticity (*Figure 5A*).

In our model, MTs are flexible polymers and their growth dynamics are described by two states. In the growth state, fibers elongate with a force-dependent growth speed. In the shrinkage state, fibers shorten with a constant shrinkage speed. The growth speed decreases exponentially with the force exerted on the fiber's tip, termed antagonistic force (*Dogterom and Yurke, 1997*). To describe the transition from growth to shrinkage, we distinguished two models: a basic 'non-pausing' model with a force-independent catastrophe rate, and a 'KIF21B-pausing' model with a force-dependent catastrophe rate (*Janson et al., 2003*) and KIF21B molecules that induce a paused state before the MT shrinks. To model control or KIF21B-KO cells, the number of KIF21B molecules can be changed. In our simulations, KIF21B motors freely diffuse in the cytosol, can bind to MTs and walk along them toward the plus end. When the motor arrives at a growing plus end, it pauses the MT dynamics until it unbinds. After a KIF21B motor has detached from the tip of a MT, the MT switches to the shrinkage state (*Figure 5B*).

To estimate numerical values for the parameters describing MT dynamics in our system, we used the experimental data for the KIF21B-KO cells (*Figure 4A–F*, *Appendix 1—table 1*). In this condition, it is reasonable to assume that every catastrophe and pause is triggered by force exerted on the MT tip, and not by KIF21B because it is absent (*Figure 5—figure supplement 1A–D*). Additionally, the KIF21B-induced mean pause time of a MT tip was set to 8.3 s, as experimentally determined (*Figure 4—figure supplement 1A*). For the remaining parameters of our simulation that were unrelated to MT growth dynamics, we used numerical values found in the literature (see *Appendix 1—table 1*).

Simulations of the non-pausing model with the force-independent catastrophe rate implied that increasing the catastrophe rate decreased the mean length of the MTs (*Figure 5C*). The mean MT length is a measure for MT density in the simulations, as we kept the cell volume and number of MTs constant. The mean MT length reached a steady state within 300 s and decreased more than threefold in the range of values for the catastrophe rate we chose (*Figure 5D*). In the KIF21B-pausing model the fibers grew to a dense MT network in the absence of KIF21B with the force-dependent catastrophe rate matched to the KIF21B-KO (*Figure 5E*). When we added KIF21B motors to mimic the control condition, MT length decreased with an increasing number of KIF21B motors (*Figure 5F*). This decrease was approximately twofold between the KIF21B-KO-like and the control-like simulations between 0 and 10 KIF21B motors.

These simulations confirmed that by changing the catastrophe rate either directly or by explicitly modeling the behavior of the pause-inducing KIF21B motor, the average length of the MTs can be regulated. Interestingly, in our simulations, a small number of KIF21B motors was sufficient to prevent the overgrowth of the MT network.

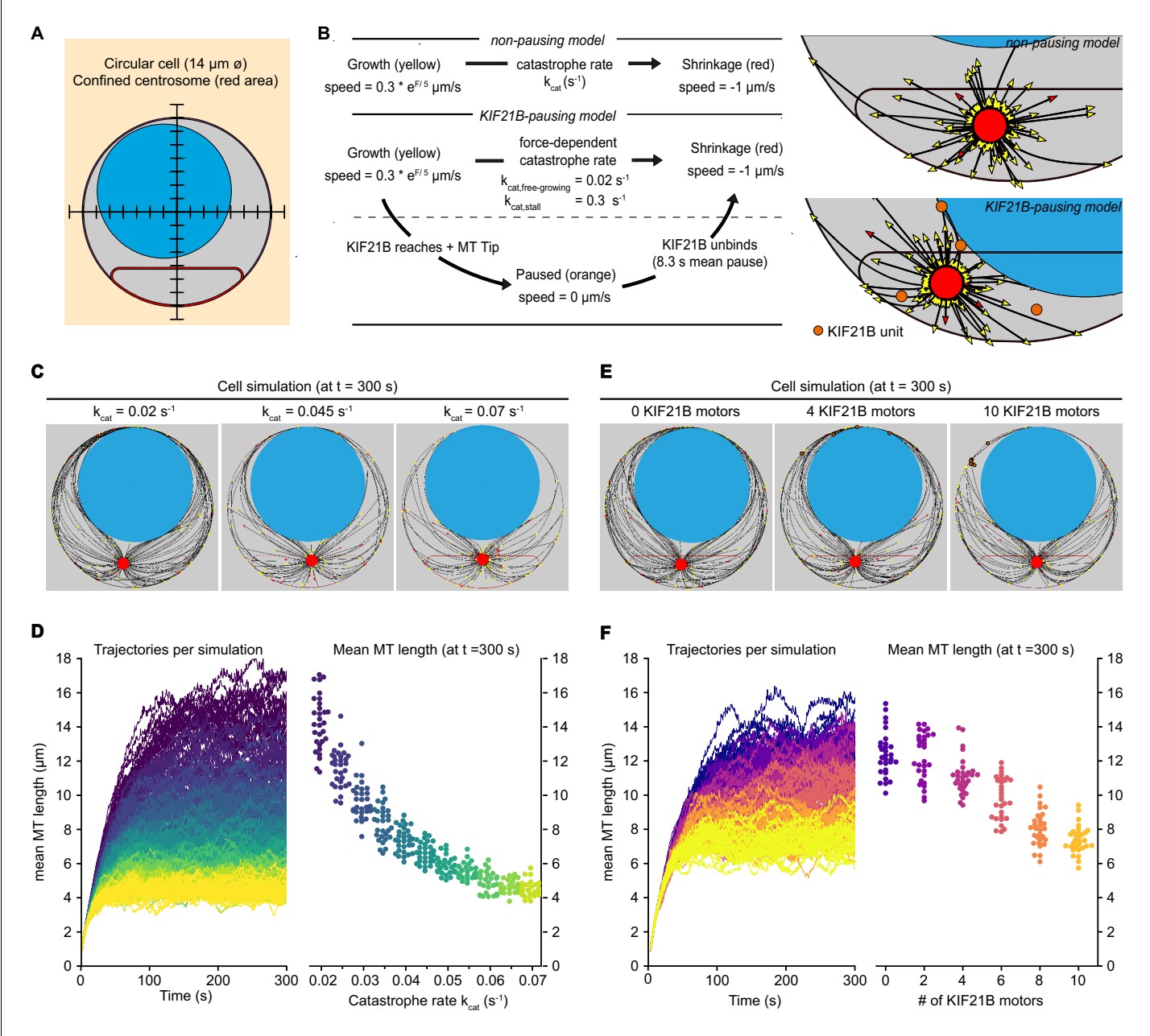

**Figure 5.** Biophysical simulation shows how KIF21B affects MT network length. (**A**) A T cell is modeled as a circular cell with a nucleus (light blue) and centrosome confinement space (red). The cell is 14 μm in diameter, the nucleus is 10 μm in diameter. (**B**) Description of modeled microtubule dynamics using two models: a non-pausing model with MTs undergoing force-dependent growth and transitioning to depolymerization stochastically with a constant catastrophe rate, and a KIF21B-pausing model, in which MT growth and catastrophe are dependent on the antagonistic force exerted on the fiber's tip. Cropped snapshots of the initial state of the non-pausing model (left, catastrophe rate = 0.045 s$^{-1}$) and KIF21B-pausing model (right, 10 KIF21B motors) are shown. Growing MT plus ends are indicated by yellow arrowheads, depolymerizing MT plus ends are indicated by red arrowheads, and KIF21B units by orange circles. (**C**) Snapshots of the non-pausing model after 300 s for different values of catastrophe rate: 0.02 s$^{-1}$ (left), 0.045 s$^{-1}$ (middle), and 0.07 s$^{-1}$ (right). (**D**) Trajectories (left) showing mean MT length per simulated cell over a 300 s time period. Traces are colored according to the catastrophe rate (from $k_{cat}$ = 0.02 to 0.07 s$^{-1}$). Quantification (right) displaying mean MT length per simulated cell at steady state (t = 300 s) for indicated values of the catastrophe rate ($k_{cat}$). n = 30 simulated cells per condition. (**E**) Snapshots of the KIF21B-pausing model after 300 s for different numbers of KIF21B motors: 0 (left), 4 (middle), and 10 (right). (**F**) Trajectories (left) showing mean MT length per simulated cell over a 300 s time period. Traces are colored according to the number of KIF21B motors present (ranging from 0 to 10 motors). Quantification (right) displaying mean MT length per simulated cell at steady state (t = 300 s) for indicated number of KIF21B motors. n = 30 simulated cells per condition.

The online version of this article includes the following source data and figure supplement(s) for figure 5:

*Figure 5 continued on next page*

*Figure 5 continued*

**Source data 1.** A CSV file with numerical data of represented trajectories as plotted in *Figure 5D*, left panel.

**Source data 2.** A CSV file with numerical data of mean MT length at time t = 300 s as plotted in *Figure 5D*, right panel.

**Source data 3.** A CSV file with numerical data of represented trajectories as plotted in *Figure 5F*, left panel.

**Source data 4.** A CSV file with numerical data of mean MT length at time t = 300 s as plotted in *Figure 5F*, right panel.

**Figure supplement 1.** Characterization of T cell force-dependent catastrophe.

**Figure supplement 1—source data 1.** A CSV file with numerical data of the fraction of MT surviving as a function of time.

**Figure supplement 1—source data 2.** A CSV file with numerical data of MT surviving under force as a function of catastrophe rates for different MT forces exerted by polymerization.

**Figure supplement 1—source data 3.** A CSV file with numerical data of the free catastrophe rate as a function of the growing force, used to obtain a linear fit and represented in *Figure 5—figure supplement 1D*.

## Restriction of MT length helps to avoid a force balance that inhibits polarization

To investigate how an overgrown MT network impairs centrosome polarization and whether KIF21B can rescue this defect, we extended our simulations to describe polarization. We converted our steady-state MT dynamics models to a polarizing system by releasing the centrosome confinement and initializing a 'synapse'. The synapse consists of a line with curved ends to which 50 dynein motors are attached (*Figure 6A*). Dynein motors are uniformly distributed over the synapse. They can bind to a MT in their vicinity and walk along them toward the centrosome. Their walking and unbinding dynamics were calculated in a force-dependent manner. Because dynein motors are anchored, walking along a MT generates a pulling force. Consequently, the centrosome is able to polarize (*Figure 6B*, *Figure 6—videos 1*, *2*).

In both the non-pausing as well as in the KIF21B-pausing model, the cells with longer MTs needed more time on average to polarize (*Figures 5D,F* and *6C,D*). Some KIF21B-KO-like cells without KIF21B and some cells with a fixed catastrophe rate smaller than 0.03 s$^{-1}$ did not polarize at the end of the simulation (800 s modeled time that we simulated after the MT network reached steady state).

We next analyzed the relation between MT length and polarization dynamics in the KIF21B-pausing model, as this model describes our experimental data. Tracking the time evolution of the centrosome-synapse distance for multiple runs provided insight into the dynamics of centrosome translocation (*Figure 6E*). The centrosomes of late polarizing cells did not travel more slowly, but rather were localized at the far opposite side of the synapse for most of the time. Once these centrosomes left this position, they quickly moved toward the synapse. This rapid polarization movement seemed independent of the number of KIF21B molecules. However, the trapped localization before the rapid repositioning indicated that the system can be relatively stable in an unpolarized state. This state can apparently be destabilized by decreasing MT length through an increasing number of KIF21B molecules.

For our simulated cells to successfully polarize, the centrosome must travel past the nucleus on either the left or the right side. A possible explanation for the stable unpolarized state may be that the forces that dynein exerts to the left and the right are equal, thus causing a mechanical equilibrium by force balance. To trigger polarization, this equilibrium needs to be broken. We quantified force imbalance by determining all forces exerted by dynein projected on the horizontal axis and sorting them according to their direction. We calculated the sum of forces pointing to the right and the sum of forces pointing to the left. For any timestep, the absolute difference of these two sums, normalized by the total sum of these forces indicates the force imbalance (*Figure 6F*, the non-normalized sum of force per timestep is shown in *Figure 6—figure supplement 1*). The resulting data revealed that dynein forces were more balanced in the simulations with low KIF21B numbers and impaired polarization than in the simulations where the number of KIF21B molecules was high and polarization occurred rapidly.

To understand the underlying mechanism, we counted the number of MTs that pass the nucleus on either the right ($MT_R$) or the left side ($MT_L$) and are bound by dynein (*Figure 6G*). We only assessed the first 10 s after initiation of polarization to limit the situation to a period when none of the runs are already actively polarizing. This analysis indicated that with increasing number of KIF21B molecules, the mean number of MTs bound on the left or on right decreases, while the difference

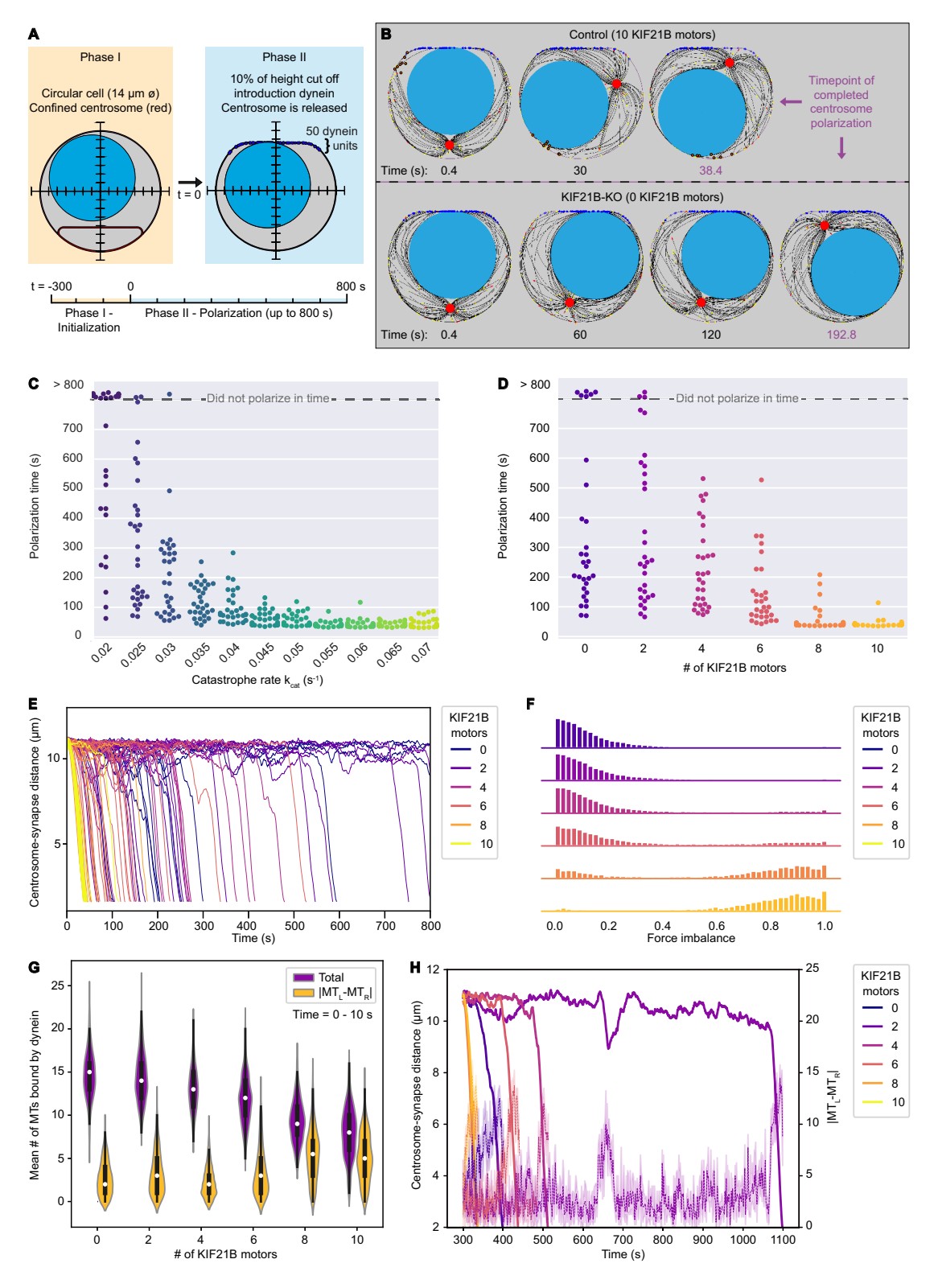

**Figure 6.** KIF21B prevents centrosome stalling induced by force balance during polarization by limiting MT length. (**A**) The T cell model is initialized to a steady state MT network in 300 s (Phase I) and is then extended to a polarizing model (Phase II). In this transition, the cell shape is changed to include a flat side with curved corners with 50 dynein units attached, and the centrosome confinement (red) is released to allow centrosome movement and translocation. (**B**) Snapshots of modeled T cells during centrosome polarization in the KIF21B-pausing model with the presence of 10 (top row) or 0

*Figure 6 continued on next page*

*Figure 6 continued*

(bottom row) KIF21B motors. The simulations were stopped after the centrosomes reached a polarized position <2.5 μm away from the synapse. The final snapshot in each row indicates the frame of the simulation when the centrosome was fully polarized. Timepoints are indicated below each snapshot. (C) Quantification of polarization time (the time the centrosome needs in Phase II to reach a polarization position) per simulated cell for the non-pausing model at indicated catastrophe rates. Some cells did not polarize within the maximum simulation time (800 s) as indicated in the upper part of the graph. n = 30 simulated cells per condition. (D) Quantification of polarization time (the time the centrosome needs in Phase II to reach a polarization position) per simulated cell for the KIF21B-pausing model with indicated numbers of KIF21B motors present. Some cells did not polarize within the maximum simulation time (800 s) as indicated in the upper part of the graph. n = 30 simulated cells per condition. (E) Trajectories showing the centrosome-synapse distance during Phase II over time along the vertical axis of the simulated T cell. Trajectories are color coded for the number of KIF21B motors present (see legend). Shown are 15 runs per condition. (F) Quantification of force imbalance within the KIF21B-pausing model. Force imbalance is calculated from the forces experienced by dynein at a given time point, projected on the horizontal axis. Force imbalance was characterized as the absolute difference of projected forces pointing to the left and forces pointing to the right, normalized by the sum of all forces. This quantity is determined for each timestep between 0 and the polarization time for every repeat and binned to create the according histogram. Histograms are color coded for the number of KIF21B motors present in the simulation (see legend). n = 30 simulated cells per condition. (G) Quantification of the mean number of MTs that are bound by dynein to the synapse within the KIF21B-pausing model. Violin plot distributions show the total number of MTs bound (purple), and the absolute difference ($|MT_L-MT_R|$, yellow) between the number of MTs bound on the right versus left side of the cell's vertical axis. Within each violin, a boxplot indicates the quartiles (black) with the inner quartile range (25–75%) indicated with the thickened region. Median values are indicated for each violin (white dot). Values were obtained in the first 10 s of Phase II of each simulation. n = 30 simulated cells per condition. (H) Trajectories of centrosome-synapse distance over time (solid lines) plotted together with trajectories of the absolute difference $|MT_L-MT_R|$ between the number of MT bound to dynein and passing the nucleus on the left or on the right (dashed lines) in the same color for the corresponding runs. To make the $|MT_L-MT_R|$ more legible, the mean and 95% confidence interval is plotted averaged over three frames (1.2 s). Trajectories are color coded for the number of KIF21B motors present in the simulation (see legend).

The online version of this article includes the following video, source data, and figure supplement(s) for figure 6:

**Source data 1.** A CSV file with numerical data of polarization time as a function of catastrophe rates represented in *Figure 6C*.
**Source data 2.** A CSV file with numerical data of polarization time as a function of the numbers of KIF21B motors represented in *Figure 6D*.
**Source data 3.** A CSV file with numerical data of the distance from the centrosome to the synapse as a function of time for different numbers of KIF21B motors, shown in *Figure 6E*.
**Source data 4.** A CSV file with numerical data of force imbalance per time trace and per time point for different numbers of KIF21B motors, as plotted in *Figure 6F*.
**Source data 5.** A CSV file with numerical data of the mean numbers of MT bound to dynein for different numbers of KIF21B motors, as shown in *Figure 6G*.
**Source data 6.** A CSV file with numerical data of time traces of the centrosome-synapse distance and time traces of the difference between the number of MTs bound by dynein passing along the right and the left side of the nucleus.
**Figure supplement 1.** KIF21B affects the balance of force in T cell polarization.
**Figure supplement 1—source data 1.** A CSV file with numerical data of summed horizontal forces on dynein per time trace and per time point for different numbers of KIF21B motors, as shown in *Figure 6—figure supplement 1*.
**Figure 6—video 1.** Recording of a simulated T cell in the 'non-pausing' model with 10 KIF21B motors added.
https://elifesciences.org/articles/62876#fig6video1
**Figure 6—video 2.** Recording of a simulated T cell in the 'non-pausing' model with 0 KIF21B motors added.
https://elifesciences.org/articles/62876#fig6video2

between the two fractions averaged over time increases. Tracking this difference as a function of time indicated that it changed together with the centrosome position, and an increased fluctuation in this number seemed necessary to trigger polarization (*Figure 6H*). Note that the dynamics of KIF21B together with the MTs and the spatial restriction by the nucleus can lead to enrichment of KIF21B on one side of the nucleus (*Figure 6B*). This specific localization could increase the difference between the number of MTs on the left and right side of the nucleus that reach the synapse and enhance the force imbalance.

We conclude that by effectively increasing catastrophe frequency and restricting MT length, KIF21B reduces the overall number of MTs reaching the synapse from different sides of the nucleus. A smaller number of long MTs leads to larger fluctuations in the number of MTs coming from opposing sides, which enables the force asymmetry that is necessary to trigger polarization when the centrosome is positioned behind the nucleus.

## Discussion

In this paper, we set out to investigate which aspects of the regulation of MT organization and dynamics are important during the formation of the immunological synapse and elucidate the role of kinesin-4 KIF21B in this process. MTs are typically quite short and sparse in immune cells, and we found that this property is functionally important, as MT overgrowth inhibited efficient centrosome translocation to the synapse (*Figure 7*). Furthermore, our data strongly support the notion that KIF21B acts as a factor that prevents excessive MT elongation.

Previous in vitro studies reported contradictory results regarding the activity of KIF21B on MT ends. One study found that KIF21B promotes MT growth and catastrophe at relatively high levels (*Ghiretti et al., 2016*). In contrast, our own work indicated that KIF21B transiently blocks MT elongation and induces subsequent catastrophes at low levels, when only one or two KIF21B molecules are present at the MT tip, and causes prolonged MT pausing when present at a higher concentration, when multiple KIF21B motors accumulate at the plus end (*van Riel et al., 2017*). In the current study, we showed that the function of KIF21B in knockout cells can be rescued by KIF21B-GFP expressed at levels that were mildly exceeding endogenous. Importantly, imaging of KIF21B-GFP in these cells did not reveal any clear accumulations of the motor at MT plus ends. KIF21B behavior in Jurkat T cells was very similar to that observed at low nanomolar concentrations in vitro (*van Riel et al., 2017*): single KIF21B-GFP kinesin dimers walked processively along MTs, reached their ends and perturbed MT plus-end growth by triggering transient pauses followed by catastrophes. Therefore, we conclude that although biochemically KIF21B can be described as a MT pausing factor, in T cells, it acts at concentrations that are insufficient to induce prolonged pausing and effectively functions as a catastrophe inducer. This is likely explained by our previous observation that a single KIF21B-GFP molecule can arrest dynamics of a few protofilaments (*van Riel et al., 2017*). Blocking of even one MT protofilament can already perturb growth of the remaining ones, and this can lead to the loss of the GTP cap and trigger catastrophe (*Doodhi et al., 2016*). To induce a pause, blocking MT elongation must be accompanied by inhibition of depolymerization of protofilaments lacking a GTP cap. Since a single KIF21B dimer is unlikely to be able to stably block depolymerization of all MT protofilaments, a catastrophe eventually ensues.

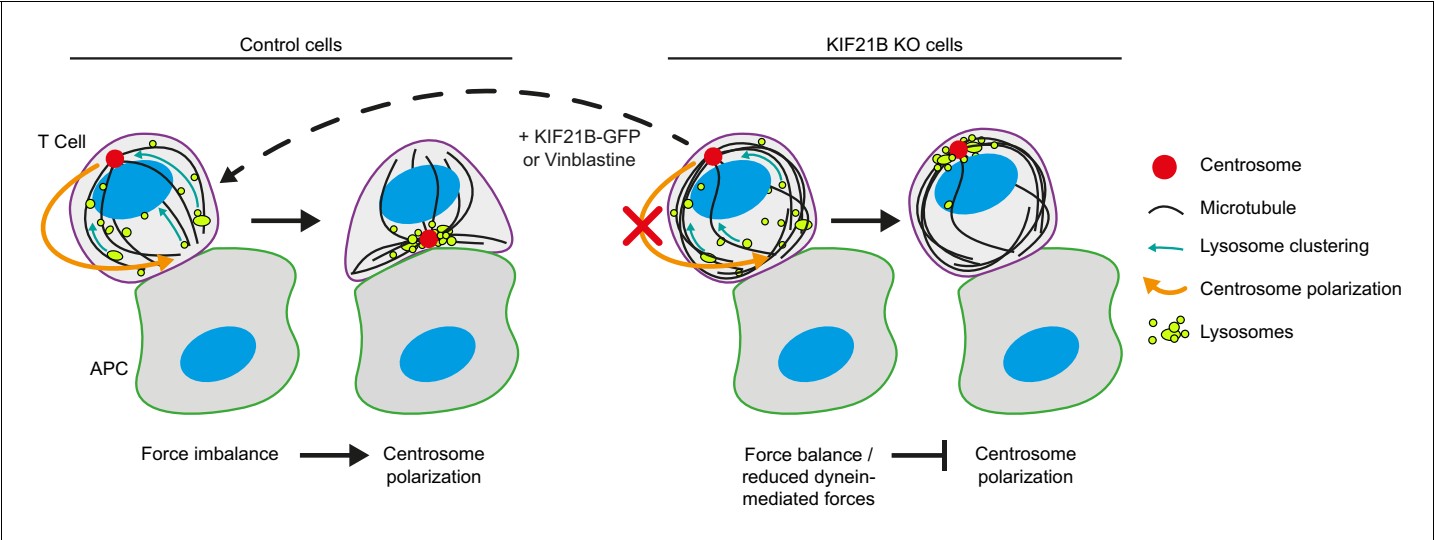

**Figure 7.** Overview of the experimental results and a model of centrosome polarization in T cells. Centrosome polarization in T cells is driven by dynein attached to the immunological synapse. KIF21B is a pausing and catastrophe-promoting factor that limits MT growth in T cells; its depletion results in MT overgrowth. Cells with overly long MTs experience difficulties in centrosome polarization. Reexpression of KIF21B-GFP or the application of a low-dose of MT-targeting agent vinblastine restores the centrosome polarization. Computational modeling suggests that an imbalance in MT pulling forces drives the centrosome movement along the nucleus. Overly long MTs create additional connections to the synapse, resulting in a balance of pulling forces on the two sides of the nucleus, thereby hindering centrosome translocation. Overly long MTs may also directly inhibit force generation by dynein at the synapse.

While KIF21B is a potent MT regulator, it has also been implicated in other functions, such as MT-based transport and regulation of signaling (*Ghiretti et al., 2016*; *Gromova et al., 2018*; *Labonté et al., 2014*; *Morikawa et al., 2018*). While our study certainly does not exclude that KIF21B engages in such functions in T cells, we show that centrosome polarization defects observed in KIF21B knockout cells can be rescued by mildly perturbing MT growth with a MT-destabilizing drug. These data strongly suggest that the primary defect induced by the lack of KIF21B is excessive MT elongation. These data also argue against direct participation of KIF21B in dynein-driven capture and shrinkage of MT ends that was proposed to drive centrosome movement in T cells (*Yi et al., 2013*), although they do not exclude indirect effects of overly long MTs on this process. Furthermore, since KIF21B is a highly processive motor, it can be expected that it would target longer MTs more effectively, like it has been previously shown for kinesin-8 (*Gupta et al., 2006*; *Varga et al., 2006*). This would make KIF21B even more efficient in preventing excessive MT growth in T cells. The finding that KIF21B acts as an inhibitor of MT growth is in agreement with the studies in neurons, which showed that KIF21B reduces MT growth processivity (*Ghiretti et al., 2016*; *Muhia et al., 2016*).

To further substantiate the idea that KIF21B affects centrosome repositioning in T cells by controlling MT length, we turned to modeling. Our simulations predict that a relatively small number of KIF21B molecules is sufficient to restrict MT length. The attachment of a KIF21B molecule to the tip of a MT induces a MT pause. After this pause, the MT switches to shrinkage, and the KIF21B motor detaches. While the MT shrinks, the KIF21B molecule can bind to the next growing MT. Because of this dynamic process of binding to, inducing shrinkage of, and unbinding from MTs, 10 KIF21B motors can, in principle, induce shrinkage of 100 MTs within 100 s. In the model, we assumed that all KIF21B molecules induce a catastrophe, which differs from our experimental findings (*Figure 3H*). However, even if only half of the KIF21B molecules induce catastrophe, as in our experimental data, the total number of KIF21B molecules needed to significantly inhibit MT overelongation would still be relatively small, and this would also likely be true even in a three-dimensional system, where the diffusional space for the KIF21B molecules would be larger and the probability to bind to a MT lower. This length regulation by only a small number of KIF21B molecules could be validated experimentally by different labeling strategies or in a reconstituted system in vitro.

Furthermore, the simulations provided a possible mechanistic explanation for the impairment of centrosome polarization by MT overgrowth. In cells where the nucleus is located between the centrosome and the synapse, as would often be the case in T cells migrating toward a chemoattractant (*Roig-Martinez et al., 2019*), the centrosome needs to pass it along one side to move to the synapse. This movement would require a force only on one side and thus an asymmetric organization of MTs reaching the synapse from the different sides of the nucleus. Such asymmetries likely originate from fluctuations in the number of MTs contacting the synapse from different sides that occur more readily when the number of MTs is low. The fluctuation-based mechanism of asymmetry generation could explain why some KIF21B knockout cells do polarize and some fail to do so within the observation period.

Although this mechanism for the impaired centrosome polarization in the absence of KIF21B appears plausible, its generalization could be limited by our model assumptions. To reduce complexity, we based our simulations on the force generation by dynein through MT sliding rather than through a MT end-on capture-shrinkage mechanism (*Yi et al., 2013*). Our model suggests that movement of the centrosome during polarization is a direct effect of a non-zero force along the MT network on one side of the nucleus. The mechanism of how this force is generated does not change the reliance on the force imbalance along the nucleus. It is possible that the precise trajectory of the centrosome depends on the molecular mechanism of how dynein generates the force (*Hornak and Rieger, 2020*). A comparison between experimentally determined centrosome trajectories and simulation results is beyond the scope of this study, and thus we cannot resolve the precise force generation mechanism of dynein. Our simulations suggest that when centrosome translocation is impaired, the MT network is experiencing balanced forces. As a consequence, we predict that in these situations one would observe major deformations of the nucleus, because it is trapped in a contracting cage of MTs spanning between the centrosome and the synapse. These deformations could also allow the centrosome to be located half-way between an apical and a basal position of the cell (*Figure 4H*). In our simulations, we assume a relatively stiff nucleus and therefore we only find the centrosome in an apical or basal position. It could be also possible that nuclear deformations push

MTs toward the synapse, where they form dense peripheral MT bundles to accommodate the least curvature (*Figure 2A and B*).

A necessary assumption to set up a computable *Cytosim* simulation was the description of a T cell as a two-dimensional system. Because of this dimensionality reduction, it is not possible to quantitatively compare all experimental results to the results of our simulations. For example, the average MT length as experimentally determined in the 3D system exceeds by approximately twofold the average MT length determined in our 2D simulations. Constraining the average MT lengths from the experiments in our 2D simulations would lead to an overcrowded system that is difficult to solve computationally. Therefore, our approach to calibrate the model to the measured MT dynamics seems the most reasonable. It is possible that in a three-dimensional system, the locked centrosome state is more unstable than we conclude from our two-dimensional simulations, because the third dimension provides an additional degree of freedom to break the force balance. However, a previous computational model in three dimensions suggested that in a T cell with a static overgrown MT network the centrosome can be trapped behind the nucleus (*Kim and Maly, 2009*), and in our ExM data we do observe KIF21B-KO cells with an apically located centrosome and symmetric MT bundles extending to the synapse from different sides (*Figure 2—figure supplement 1B*). It is important to note that in other cells in our ExM samples the centrosome is located on the side of the nucleus, a situation that would lead to rapid polarization in our simulations. Since a significant proportion of cells shows a polarization delay in our experiments, it is unlikely that trapping of the centrosome behind the nucleus is the sole explanation of the polarization defect. It is possible that also in situations where the centrosome is positioned on the side of the nucleus, MT overgrowth leads to force balance and prevents efficient centrosome translocation. Furthermore, other mechanisms that are not included in the simulations likely contribute to the inhibition of centrosome repositioning by overgrown MTs. Excessively long MTs could slow down centrosome translocation by directly affecting dynein-mediated force generation, for example, by reducing the number of MT ends contacting the synapse and perturbing MT end capture by dynein. Long MTs could also make MT depolymerization-driven 'reeling in' of the centrosome less efficient or create pushing forces and friction that would inhibit repositioning of the MT network. We further note that even in cells that did relocate their centrosome, the immunological synapses formed slower and were smaller, suggesting that the presence of circular MT bundles might perturb synapse formation in some additional ways, for example, by affecting signaling-based cross-talk between MTs and actin (*Dogterom and Koenderink, 2019*).

Taken together, our data show that the typical features of MT network organization in T cells, with relatively short and sparse MTs, is functionally important and that its maintenance depends on a dedicated regulatory factor, KIF21B, which restricts MT growth.

## Materials and methods

The information on key resources can be found in Appendix 1 - Key Resources Table.

### Cell culture, spreading assays, and drug treatment

Jurkat T cells (clone E6.1; ATCC TIB-152) were grown in RPMI 1640 medium w/L Glutamine (Lonza) supplemented with 10% Fetal Bovine Serum and 1% penicillin/streptomycin. For all spreading assays, coverslips were coated with poly-D-lysine (Thermo Fisher Scientific, A3890401), washed with phosphate buffered saline (PBS) and incubated overnight at 4°C with a mouse monoclonal anti-CD3 antibody (clone UCHT1, StemCell Technologies, #60011) or a mouse monoclonal anti-HA antibody (clone 16B12, Biolegend (Covance), MMS-101P) at 10 μg/mL in PBS, except for the not-activated conditions, here cells were incubated on poly-D-lysine-coated coverslips for 10 or 30 min prior to fixation. For live-cell imaging, Lab-Tek chambers (Thermo Fisher Scientific, 155409) were incubated overnight at 4°C with anti-CD3 (Clone UCHT1, StemCell Technologies, #60011) 10 μg/mL in PBS. Prior to spreading, cells were spun down for 4 min at 1000 rpm and resuspended in fresh prewarmed RPMI 1640 medium.

Vinblastine (Sigma-Aldrich) treatment was performed by spinning down the cells for 4 min at 1000 rpm and resuspending the cells in pre-warmed medium containing 0.5 nM vinblastine 30 min prior to fixation. For live-cell imaging experiments, cells were resuspended in medium and 30 min

prior to imaging, an equal amount of medium with 1 nM vinblastine was added to the cells to achieve a final concentration of 0.5 nM vinblastine.

Cell lines were routinely checked for mycoplasma contamination (LT07-518 Mycoalert assay, Lonza).

## CRIPSR/Cas9 knockouts, lentivirus transduction, and DNA constructs

KIF21B knock-out cell lines were generated in Jurkat cells (clone E6.1) using CRISPR/Cas9 technology. Jurkat T cells were transfected using an Amaxa Cell Line Nucleofector kit V (Lonza), program X-005 with the pSpCas9(BB)−2A-Puro (PX459) vector (Addgene, #62988) bearing the appropriate targeting sequence (KIF21B: 5'-caccgTGTGTGAGCAAGCTCATCGA-3'). Cells were selected using 2 µg/mL puromycin (InvivoGen).

cDNA for lentivirus constructs were derived from the following sources: human cDNA clone KIAA 0449 for KIF21B (Kazusa DNA Research Institute), the EB3-GFP coding sequence as described in *Stepanova et al., 2003*, and the β-tubulin coding sequence was provided by Prof. Dr. Kai Jiang. KIF21B-GFP and β-tubulin-GFP sequences were cloned in pLVX-IRES-Puro vectors (Clontech), EB3-GFP was cloned in pLVX-IRES-Hygro vector (Clontech). EB3-mCherry construct (*Stepanova et al., 2003*) was transiently transfected using Amaxa Cell Line Nucleofector kit V (Lonza), program X-001.

Lentiviruses were produced by MaxPEI-based transfection of HEK293T (ATCC CRL-11268) cells with the construct of interest and the packaging vectors psPAX2 and pMD2.G (Addgene). Cell supernatant was harvested 48 and 72 hr after transfection, filtered through a 0.45 µm filter and incubated overnight at 4°C in a polyethylene glycol (PEG) 6000-based precipitation buffer containing 80 g/L PEG 6000 and 82 mM NaCl at pH 7.2. The precipitation mix was centrifuged to concentrate the virus. The pellet containing the lentivirus was resuspended in PBS and Jurkat T cells were transduced in complete medium supplemented with 8 µg/mL polybrene (Merck-Millipore, TR-1003-G) using a spinoculation protocol: centrifugation of the transduction mix for 30 min at 2400 rpm at 32°C. Cells were selected using 2 µg/mL puromycin (InvivoGen, ant-pr5b) or with 100 µg/mL hygromycin (Invivo-Gen, ant-hm), five days after transduction.

## Western blotting and antibodies

Jurkat T cells were lysed in a lysis buffer containing 20 mM Tris (pH 7.5), 150 mM NaCl, 1% Triton X-100, and cOmplete protease inhibitor cocktail (Roche, 4693132001). For Western blotting, we used the following polyclonal rabbit antibodies: anti-KIF21B (Sigma-Aldrich, HPA027249), anti-GFP (Abcam, ab290); and the following mouse monoclonal antibodies: anti-CD3 (Clone UCHT1, StemCell Technologies, #60011), and anti-Ku80 (BD Bioscience, 611360). The following secondary antibodies were used for Western blotting: IRDye-800CW-conjugated goat antibody against mouse IgG (#P/N 925–32210) and IRDye-680LT-conjugated goat antibody against rabbit IgG (#P/N 925–68021), both purchased from Li-Cor Biosciences.

For immunofluorescence staining, we used a rabbit polyclonal antibody against CEP135 (Sigma-Aldrich, SAB4503685). A rabbit monoclonal antibody against Lamtor4 (clone D6A4V, Cell Signaling Technology, #12284) and α-tubulin (clone EP1332Y, Abcam, ab52866). A mouse monoclonal antibody against α-tubulin (Sigma-Aldrich, T6199), and a rat monoclonal antibody against α-tubulin (clone γL1/2, Abcam, ab6160). The following secondary antibodies were used for immunofluorescence: Alexa-Fluor-488-conjugated goat antibodies against mouse and rabbit (#A27023 and #A-11034), Alexa-Fluor-594-conjugated goat antibodies against mouse, rabbit, and rat (#A-11032, #R37117 and #A-11007) and an Alexa-Fluor-647-conjugated antibody against mouse (#A-21240); all these antibodies were purchased from Life Technologies. F-actin was stained using Alexa-Fluor-488 or 594-conjugated phalloidin (Life Technologies, #12379, #12381).

## T cell stimulation, RNA isolation, and qRT-PCR analysis

T cells were stimulated with a combination of Phorbol 12-myristate 13-acetate (PMA) (Sigma-Aldrich, P8139) and ionomycin (Sigma-Aldrich, I0634) at indicated time points and concentrations. Cells were collected on ice and total RNA was extracted using TRIzol reagent (Thermo Fisher Scientific, 15596026) and cDNA synthesis was performed using the iScript cDNA synthesis kit (Bio-Rad, 1708891). cDNA samples were amplified with SYBR Select mastermix (Life Technologies, 44-729-19) using a QuantStudio 12K Flex Real-Time PCR System (Applied Biosystems) and the following RT-

PCR primer pairs: IL-2 forward 5'-AACTCACCAGGATGCTCACATTTA-3', IL-2 reverse 5'-TCCC TGGGTCTTAAGTGAAAGTTT-3' and GAPDH forward 5'-CAACGGATTTGGTCGTATT-3' and GAPDH reverse 5'- GATGGCAACAATATCCACTT-3'. The LIVAK method was used to calculate relative mRNA expression in respect to the housekeeping gene GAPDH.

## Immunofluorescence staining and image acquisition

For immunofluorescence staining of tubulin and CEP135 for TIRF microscopy, cells were fixed in −20℃ methanol for 10 min, followed by washes with PBS and permeabilization with PBS supplemented with 0.15% Triton X-100 for 2 min. All subsequent wash steps were performed using PBS supplemented with 0.05% Tween-20. Epitope blocking and antibody labeling steps were performed in PBS supplemented with 0.05% Tween-20% and 1% Bovine Serum Albumin (BSA). Before mounting in DAPI-containing Vectashield mounting medium (Vector Laboratories, H-1200–10), slides were washed with 70% and 100% ethanol and air-dried.

For immunofluorescence staining of lysosomes (Lamtor4) and centrosomes (CEP135) for 3D analysis, or labeling with Phalloidin, cells were fixed with 4% paraformaldehyde (PFA) in PBS at room temperature (RT) for 10 min, followed by washes with PBS and permeabilization with PBS supplemented with 0.15% Triton X-100 for 2 min. All subsequent wash steps were performed using PBS supplemented with 0.05% Tween-20. Epitope blocking and antibody labeling steps were performed in PBS supplemented with 0.05% Tween-20% and 1% BSA. Samples were mounted using DAPI-containing Vectashield mounting medium (Vector Laboratories, H1200-10).

Widefield imaging on fixed cells was performed on a Nikon Eclipse Ni-E upright fluorescence microscope equipped with Plan Apo Lambda 100x N.A. 1.45 oil and 60x N.A. 1.40 oil objectives microscopes, ET-BFP2 (49021, Chroma), ET-GFP (49002, Chroma), ET-mCherry (49008, Chroma), ET-Cy5 (49006, Chroma) filter sets and a Photometrics CoolSNAP HQ2 CCD (Roper Scientific, Trenton, NJ) camera. The microscope was controlled by Nikon NIS Br software.

TIRF imaging on fixed cells was performed on an ILAS-2 system (Roper Scientific, Evry, France) with a dual laser illuminator for azimuthal spinning TIRF (or Hilo) illumination and with a custom modification for targeted photomanipulation. This system was installed on Nikon Ti microscope (with the perfect focus system, Nikon), equipped with 405 nm 100 mW Stradus (Voltran), 488 nm 150 mW Stradus (Voltran), 561 nm 100 mW Coherent (OBIS 561-100LS) and 642 nm 110 mW Stradus (Vortran) lasers; and ET-BFP (49021, Chroma), ET-GFP (49002, Chroma) and ET-mCherry (49008, Chroma) filter sets. Fluorescence signal was detected using an EMCCD Evolve mono FW DELTA 512 × 512 camera (Roper Scientific) with the intermediate lens 2.5X (Nikon C mount adapter 2.5X) or a CCD CoolSNAP MYO M-USB-14-AC camera (Roper Scientific). The final resolution using EMCCD camera was 0.065 µm/pixel, using CCD camera it was 0.045 µm/pixel. The setup was controlled with MetaMorph 7.8.8 software (Molecular Device).

Spinning disk microscopy was performed on inverted research microscope Nikon Eclipse Ti-E (Nikon), equipped with the perfect focus system (Nikon), Plan Apo VC 100x N.A. 1.40 oil objective (Nikon), spinning disk Yokogawa CSU-X1-A1 with 405-491-561-642 quad-band mirror (Yokogawa). The system was also equipped with ASI motorized stage with the piezo plate MS-2000-XYZ (ASI), Back-Illuminated Evolve 512 EMCCD camera (Photometrics) or Back-Illuminated Prime BSI sCMOS camera (Photometrics) and controlled by the MetaMorph 7.10 software (Molecular Devices). Here, 405 nm 100 mW Stradus (Voltran), 491 nm 100 mW Calypso (Cobolt), 561 nm 100 mW Jive (Cobolt), and 642 nm 110 mW Stradus (Vortran) lasers were used as the light sources. The setting was equipped with ET-BFP2 (49021, Chroma), ET-GFP (49002, Chroma), ET-mCherry (49008, Chroma) and ET-Cy5 (49006, Chroma) filter sets to image blue, green, red and far-red fluorescence signals, respectively. 16-bit images were projected onto the Evolve 512 EMCCD camera with intermediate lens 2.0X (Edmund Optics) at a magnification of 0.066 µm/pixel or onto Prime BSI sCMOS camera with no intermediate lens at a magnification of 0.063 µm/pixel. Z-stacks of cells were acquired with a step size of 0.1 µm.

## Immunofluorescence staining and image acquisition for STED microscopy and ExM

For all STED and ExM samples, 1.5-mm-thick coverslips (Marienfeld, 0107032) were used. Jurkat cells were added to anti-CD3 coated coverslips for indicated time points. Cells were pre-extracted

for 1 min with pre-warmed (37°C) extraction buffer composed of MRB80 (80 mM K-PIPES pH 6.8, 4 mM MgCl₂, 1 mM EGTA) supplemented with 0.35% Triton X-100% and 0.2% glutaraldehyde. After extraction, cells were fixed for 15 min with pre-warmed (37°C) 4% PFA in PBS. All subsequent wash steps were performed using PBS supplemented with 0.2% Triton X-100. Epitope blocking and antibody labeling steps were performed in PBS supplemented with 3% BSA. Labeling with primary antibodies was performed overnight at 4°C. After washing with PBS, labeling with secondary antibodies was performed for 3 hr at RT. For STED microscopy samples, cells were washed with PBS and MiliQ and air dried. Cells were mounted in Prolong Gold (Thermo Fisher).

For ExM, we followed the procedures described by *Jurriens et al., 2020*. Samples were post-fixed with 0.1 mg/mL acryloyl X-SE (AcX) (Thermo Fisher, A20770) in PBS overnight at RT. For gelation, monomer solution was prepared containing 2M NaCl, 8.625% sodium acrylate (Sigma-Aldrich, 408220), 2.5% acrylamide (AA), 0.15% N,N'-methylenebisacrylamide (BIS) in PBS. As a source of AA we used 37.5:1 AA/BIS solution (Sigma-Aldrich, A3699) and supplemented it with BIS (Sigma-Aldrich, M1533) to reach final concentrations. Gelation of the monomer solution was initiated with 0.2% ammonium persulfate (APS) and 0.2% tetramethylethylenediamine (TEMED) and 170 μL was transferred to a silicone mold with inner diameter of 13 mm (Sigma-Aldrich, GBL664107) attached to a parafilm-covered glass slide, with the sample put cell-down on top to close off the gelation chamber. After incubation at RT for 1–3 min, the sample was transferred to a humidified 37°C incubator for at least 30 min to fully polymerize the gel. After gelation, the gel was transferred to a 12-well plate and digested in TAE buffer (containing 40 mM Tris, 20 mM acetic acid and 1 mM EDTA) supplemented with 0.5% Triton X-100, 0.8 M guanidine-HCl and 7.5 U/mL Proteinase-K (Thermo Fisher, EO0491) for 4 hr at 37°C. The gel was transferred to a Petri dish, water was exchanged twice after 30 min and sample was left in water to expand overnight in 50 mL MiliQ. Prior to imaging the cells were trimmed and mounted.

All STED and ExM images have been acquired using a Leica TCS SP8 STED 3X microscope equipped with a HC PL Apo 100x/1.40 Oil STED WHITE objective for STED acquisition and a HC PL APO 86x/1.20W motCORR STED (Leica 15506333) water objective for ExM. A pulsed white laser (80 MHz) was used for excitation, and when using STED a 775 nm pulsed depletion laser was used. The internal Leica GaAsP HyD hybrid detectors were used with a time gate of $1 \leq tg \leq 6$ ns. The set-up was controlled using LAS X. If required, drift correction was performed on ExM acquisitions using Huygens Software (Scientific Volume Imaging).

## Live-cell imaging

Live-cell DIC imaging was performed on an inverted Nikon Ti microscope equipped a perfect focus system (Nikon), a Plan Fluor 40x/1.30 Oil DIC objective, a CoolSNAP HQ2 CCD camera (Photometrics), a motorized stage MS-2000-XYZ with Piezo Top Plate (ASI), and a stage top incubator (Tokai-Hit) set to 37°C. The microscope was controlled by MicroManager software.

Jurkat cells were live imaged on an ILAS-2 system (Roper Scientific, Evry, France) is a dual laser illuminator for azimuthal spinning TIRF (or Hilo) illumination and with a custom modification for targeted photomanipulation. This system was installed on Nikon Ti microscope (with the perfect focus system, Nikon), equipped with a Nikon Apo TIRF 100x N.A. 1.49 oil objective (Nikon); 488 nm 150 mW Stradus (Voltran) and 561 nm 100 mW Coherent (OBIS 561-100LS) lasers; and ET-GFP (49002, Chroma) and ET-mCherry (49008, Chroma) filter sets. For simultaneous imaging of green and red fluorescence, we used an ET-GFP/mCherry filter cube (59022, Chroma) together with an Optosplit III beamsplitter (Cairn Research Ltd) equipped with double-emission filter cube configured with ET525/50 m, ET9630/75 m, and T585lprx (Chroma). Fluorescence signal was detected using an EMCCD Evolve mono FW DELTA 512 × 512 camera (Roper Scientific) with the intermediate lens 2.5X (Nikon C mount adapter 2.5X) or a CCD CoolSNAP MYO M-USB-14-AC camera (Roper Scientific) The setup was controlled with MetaMorph 7.8.8 software (Molecular Device). To keep cells at 37°C, a stage top incubator model INUBG2E-ZILCS (Tokai Hit) was used. Or on an inverted research microscope Nikon Eclipse Ti-E (Nikon) with the perfect focus system (PFS) (Nikon), equipped with Nikon CFI Apo TIRF 100 × 1.49 N.A. oil objective (Nikon). For excitation lasers we used 491 nm 100 mW Stradus (Vortran). We used an ET-GFP 49002 filter set (Chroma) for imaging of proteins tagged with GFP. Fluorescence signal was detected using an an Evolve512 EMCCD camera (Photometrics) or a CoolSNAP HQ2 CCD camera(Photometrics). The setup was controlled with MetaMorph 7.8.8 software

(Molecular Device). To keep cells at 37°-C, a stage top incubator model INUBG2E-ZILCS (Tokai Hit) was used.

## Image processing and analysis

Images and movies were processed using ImageJ. All images were modified by linear adjustments of brightness and contrast. Average intensity projections and z-projections were made using the z projection tool. Kymographs were made using the ImageJ plugin KymoResliceWide v.0.4 (*Katrukha, 2017*).

### Cell surface area

Cell surface area was measured manually by drawing the cell circumference using the freehand tool in ImageJ. For KIF21B-GFP overexpressing rescue cell lines, very low expressing or GFP-negative cells were not measured.

### Cell volumes

Cell volumes were measured based on Phalloidin staining of 3D-imaged Jurkat cells imaged on a Spinning disk confocal microscope. Imaris software (Bitplane/Oxford instruments, version 9.5.1) was used to manually create a cell surface mask using the Isosurface Drawing mode set at 10% density. The cell outline was determined for >10 positions in z-dimension to create a 3D volume rendering of the Jurkat cell from which the volume was determined using the Statistics feature of the Imaris software.

### MT organization (TIRF microscopy)

For quantifying MT organization in T cells in different KIF21B-KO and rescue conditions, we classified MT networks as 'radial' when a clear aster-like MT pattern was observed with an obvious presence of a MT-organizing center or CEP135-positive fluorescent signal in the TIRF field. Other MT networks were classified as 'disorganized' when an aster-like organization indicative of a polarized centrosome was lacking.

### 3D analysis of lysosomes and centrosomes

To quantify polarization of lysosome clusters in T cells, the distance of these Lamtor4-positive clusters and the anti-CD3-positive coverslip was measured in z. A reslice was made of the entire cell to have a z-projection from which an average intensity projection was made to create a landscape of fluorescence intensities in z. The distance between coverslip and lysosome cluster was determined by taking the distance between the coverslip and the pixel row in z that contained the peak intensity value of all rows. A Lamtor4-stained lysosome cluster was quantified as 'polarized' when its peak intensity value was localized within 2 µm distance from the anti-CD3-stained coverslip.

For centrosome polarization quantifications, the distance between the CEP135-stained centrosome and the anti-CD3-positive coverslip was measured in z. A line scan was drawn over the centrosome and a reslice was made to make a z-projection of the cell at the location of the centrosome. The distance between coverslip and centrosome was determined by drawing a line from the center to center of both fluorescence signals. A CEP135-stained centrosome was quantified as 'polarized' when it was localized within 2 µm distance from the anti-CD3-stained coverslip.

### STED microscopy

STED images of fluorescently labeled MTs were analyzed using ImageJ. Images were background subtracted using the Background Subtraction tool with a rolling ball radius of 50 pixels and were separated into radial and non-radial components using a customized ImageJ macro (*Katrukha, 2018*; *Martin et al., 2018*). In short, we used the 'Cubic Spline Gradient' method and Tensor Sigma parameter of 8 pixels to calculate a radial and a non-radial map image that illustrate the separated radial and non-radial components of the original picture. For both the original image and images showing the separated components, radial intensity profiles were made from the centrosome to the farthest removed portion of the cell periphery using the Radial Profile Angle plugin at a 45° integration angle. To quantify the non-radiality of the MT organization per cell, the areas under the curve (AUC) of the radial intensity profiles of the original image (total intensity) and of the non-

radial map images were calculated using Excel. The non-radial intensity as percentage of the total intensity was calculated using: $(AUC_{non-radial}/AUC_{Total\ intensity})*100$.

## Expansion microscopy

For the ExM analyses and all 3D renders Imaris (Bitplane/Oxford instruments, version 9.5.1) was used. Using the Automatic Spot Detection algorithm (set to 10 µm), the centrosome was localized in the 3D renderings as the brightest spot present; and a sphere was generated around the centrosome with a radius of 5 µm. A second sphere was generated with a radius of 6 µm using the Automatic Spot Detection (set to 11 µm). From these two spheres, a spherical shell was generated with an inner spacing thickness of 1 µm using the following steps: all voxels outside the surface of the smaller sphere and all voxels inside the surface of the bigger sphere were set to the maximum intensity value (255). Using the Colocalization function, a colocalization channel of both masks was made which generated the spherical shell used for all further quantifications.

To analyze MTs within this spherical shell, a colocalization channel was generated for the tubulin channel using the Colocalization function. Thresholds were adjusted to include all MTs. Based on this colocalization channel, surface masks were generated for the tubulin signal using the Surfaces tool with Split Touching Objects enabled (seed points diameter set at 0.45 µm). Seed points and generated surfaces were visually filtered so that all MTs were included. To count the number of MTs, all generated surfaces were checked and the number of MTs were counted manually per spherical shell. The mean intensity corresponding to 1 µm MT was determined from the fluorescence intensity per surface (using the Statistics feature of the Imaris software) corrected for the manually counted number of MTs in each surface. To determine the total fluorescence intensity corresponding to the complete MT network, the Surfaces tool was used to generate a surface mask covering the entire MT network (Split Touching Objects setting disabled) and used an automatic threshold to discard very small nonspecific surfaces. To estimate the average MT length, the total intensity as found (using the Statistics feature of the Imaris software) was divided by the MT intensity per micron and by the number of MTs.

## Quantification of live-cell imaging

The duration of synapse formation imaged using DIC microscopy was quantified using kymographs. Kymographs were generated using the ImageJ plugin KymoResliceWide v.0.4. along a line from the center of the cell to the cell periphery showing least irregular cell spreading. Cells that exhibited drift on the coverslip during spreading were not used for quantification. The initiation of synapse formation was determined as the moment where the cell started to spread sideways. The synapse was at maximum size when sideway spreading became static (velocity <0.5 µm/min). The duration of synapse formation was determined as the duration at which the Jurkat cell exhibited sideward spreading at a minimum mean velocity of 1 µm/min.

EB3-GFP growth rates were quantified using kymographs that were generated using the ImageJ plugin KymoResliceWide v.0.4. The slope of EB3-GFP growth events in these kymographs were used to calculate growth rates corrected for acquisition settings. Catastrophe rates were quantified by determining the inverse value of growth time of EB3-GFP growth events.

Single-molecule KIF21B velocities were quantified from TIRF-imaged KIF21B-GFP expressing T cells using kymographs that were generated using the ImageJ plugin KymoResliceWide v.0.4. The slope of motile KIF21B-GFP events in these kymographs were used to calculate kinesin velocities corrected for acquisition settings. These values were fitted to a Gaussian curve to determine mean KIF21B velocity.

Single kinesin events at MT plus ends were quantified using TIRF-imaged KIF21B-GFP expressing T cells that were transfected with EB3-mCherry as a marker of growing MT plus ends. Events were quantified when a motile KIF21B-GFP molecule (>0.1 µm/s velocity) reached the end of a growing EB3-mCherry-positive MT. KIF21B-GFP behavior after a MT plus-end encounter was categorized as 'pausing', 'tip-tracking' or 'dissociation'. The 'tip-tracking' events, where KIF21B tracked a growing plus end at the same velocity as EB3 signal, were subsequently sub-categorized into 'tip-tracking + pausing' and 'tip-tracking + dissociation' events. The state of the EB3-mCherry-positive plus end was defined as 'growing' (>0.05 µm/s growth velocity) or 'static' (<0.05 µm/s growth velocity) for KIF21B-GFP events that went into a paused state, regardless whether there was tip-tracking or not.

The fate of EB3-mCherry plus ends after encountering a pausing KIF21B-GFP molecule was divided into four categories: 'pausing + catastrophe', 'pausing + growth reinitiation', 'growth continuation without pausing' or 'not determined' for the events that were unclear or took place beyond acquisition time. The fate of EB3-mCherry plus ends after encountering a KIF21B-GFP molecule that directly dissociated upon reaching the plus end was divided into three categories: 'no effect', 'coincided with MT pausing' or 'coincided with MT catastrophe'. Pause times were quantified for KIF21B-GFP molecules that that appeared static (<0.1 μm/s velocity and ≥0.5 s duration) and underwent pausing at an EB3-mCherry-positive MT plus end. Pause times were quantified for growing EB3-mCherry MT plus ends that appeared static (<0.05 μm/s growth velocity and ≥0.5 s duration) and transitioned into a paused state after an encounter with a KIF21B-GFP molecule. The end time of a pause coincided with either a catastrophe or a growth re-initiation event.

For analysis of β-tubulin-GFP imaging, we scored the fate of growing MTs reaching the plasma membrane into four categories: catastrophe, sliding along the cortex, MT buckling, and pausing. Quantified events had to meet the following criteria: (1) events at the MT plus end had to be preceded by a phase of growth before reaching the plasma membrane; (2) the acquisition time had to include a complete pausing event or, in the case of a catastrophe, sliding or buckling event, the initiation of the event. Pause events and pause times were quantified for MT plus ends that appeared static (<0.05 μm/s growth velocity and ≥0.5 s duration). For determining pause duration of a MT reaching the plasma membrane, a kymograph was generated for the growing MT using the ImageJ plugin KymoResliceWide v.0.4 and the length of the pause was determined by the number of vertical pixels in the kymograph multiplied by the exposure time.

## Single-molecule intensity analysis

Single-molecule fluorescence histograms of monomeric GFP (control) or kinesins moving on MTs intracellularly were built from acquisitions made on a TIRF microscope. To ensure identical imaging conditions, a single imaging slide (with a plasma cleaned coverslip) was used containing two flow chambers to image GFP (control) and KIF21B-GFP expressing T cells. For purified GFP, we used cell lysates from HEK293T cells overexpressing monomeric GFP. The GFP protein was diluted in PBS and added to an imaging flow chamber; chambers were subsequently washed with PBS, leaving a fraction of the GFP proteins immobilized on the coverslip. Protein dilution was optimized to provide images of ~0.05 fluorophores/μm$^2$ for GFP control conditions. For T cells expressing KIF21B-GFP, the imaging chamber was incubated overnight at 4°C with 10 μg/mL of a mouse monoclonal anti-CD3 (clone UCHT1, StemCell Technologies, #60011) in PBS. The chamber was washed three times with PBS and one time with RPMI 1640 medium. T cells expressing KIF21B-GFP were concentrated in RPMI 1640 medium and added to the chamber 2 min before imaging. After sealing the chambers with vacuum grease to prevent evaporation, samples were imaged on a TIRF microscope at 37°C. For monomeric GFP, ~40 images were acquired at 100 ms exposure time at different positions on the coverslip to avoid pre-bleaching. For moving kinesins, ~5–10 cells were imaged using stream acquisition at a 100 ms exposure time per frame. All acquisitions were obtained under identical laser power and a TIRF angle with a calibrated penetration depth $d$ of 180 nm.

ImageJ plugin DoM_Utrecht v.1.1.6 was used to for detection and fitting of single molecule fluorescent spots as described previously (*Yau et al., 2014*). In short, individual spots were fitted with 2D Gaussian and the amplitude of the fitted Gaussian function was used as a measure of the fluorescence intensity value of an individual spot. For moving KIF21B-GFP molecules in T cells, ImageJ plugin DoM_Utrecht v.1.1.9 was used for detection and fitting of single molecule fluorescent spots. Using this plugin, these particles were linked to tracks with a maximum distance to search over one frame of 3 pixels and a maximum linking gap of 2 pixels. Tracks were filtered for a duration of ≥1 s. The histograms were fitted to lognormal distributions using GraphPad Prism 8. A correction factor (c) was calculated for KIF21B-GFP fitted peak intensities (i) for a series (0–100 nm) of hypothetical distances (h) between the coverslip and KIF21B molecule using the formula:

$$c = \frac{i}{e^{-\left(\frac{h}{180}\right)}}$$

### *Cytosim* modeling

The T cell model was made in the agent-based modeling structure *Cytosim* (**Nedelec and Foethke, 2007**), which solves overdamped Langevin equations to calculate the movement of the cytoskeleton and associated proteins. We choose *Cytosim* to be able to explicitly model the dynamics of KIF21B and dynein and to ensure an easy reproducibility of our results. *Cytosim* is open-source and together with our configuration file easy to run. For generating plots and figures, we used Python with Seaborn, NumPy, and Pandas libraries.

To set up a simulation of the MT network and the associated proteins of a T cell, we used the following features and elements.

### Shape

The shape of the T cell is encoded as SpaceHemisphere. This space definition makes a circular (2D) or spherical (3D) space, where a plane intersects the space at arbitrary height. In the T cell, this is at 0.9 of the full height. In the space, we use to confine the centrosome during initialization this is at 0.2 of the full height of the cell. The corners of the plane are curved, to avoid infinitesimal corners. The curved corners are calculated by finding a smaller circle that interpolates continuously between the synapse and the (circular) cell shape. Any interaction with the shape is calculated by finding the closest point on the space surface and applying a linear force in this direction. If objects are outside of the space, they are pushed inwards.

### Steric interactions

Because we simplified our T cell to a two-dimensional object, we needed to adjust the steric interactions. Fibers in the simulations cannot enter the nucleus, but can cross one another to effectively mimic the third dimension. This behavior is achieved by the 'ad-hoc' steric interactions within *Cytosim,* in which we compare the position of each fiber segment with the nucleus, and if the fiber segment is within the nucleus, a linear force is applied to push the fiber out, similar to cell shape confinement.

### Dynein

Dynein motors are uniformly distributed and anchored at the synapse. They bind to MT in their vicinity and walk along them. The walking speed is characterized by a linear force-velocity relation as typically modeled in *Cytosim*. This definition describes a cortically anchored dynein that slides MTs, which is different from the end-on capture mechanism proposed by *Yi et al., 2013*.

### KIF21B

The KIF21B units are modeled as a new form of binding Hand in *Cytosim*: the Capper hand. They are able to bind and walk on the fiber. When they reach a fiber's tip, the fiber is set to a paused state, neither growing or shrinking. The waiting time for the motor at a fiber's tip is redrawn from a distribution to ensure that the pause length is correctly implemented, regardless of how long the Capper has already walked on the fiber. If a Capper unbinds, it sets the MT to the shrinkage state. Subsequently, any other bound Cappers unbind.

As KIF21B is localized in the cytosol, we extended the ad-hoc steric interactions to Singles. Because Singles are modeled as molecules with a negligible inertia displaying a random walk, we cannot easily apply elastic forces to these objects. Thus, we model the KIF21B units to bounce off the nucleus, similar to its interaction with the shape. If a good bounce location cannot be found after 1000 attempts (i.e. the molecule is in a tight corner between nucleus and cell shape), the molecule is set to the edge of the nucleus.

### Fitting KIF21B-KO MT dynamics

To match the parameterization of our model to the MT dynamics in the KIF21B-KO system we used a separate *Cytosim* simulation to mimic the experimental data from the TIRF imaging of the MT dynamics at the synapse. Ultimately, this approach suggested a linear relation between the free growing catastrophe rate and the growing force, which is the characteristic force of the MT polymerization. This relation fulfills the constraint that 81% of the MT in the TIRF data did not initiate

catastrophe within 3.3 s, while satisfying the 0.3 s$^{-1}$ stalled catastrophe rate. More details are included in *Figure 5—figure supplement 1A–D*.

In all simulations outside of fitting the KIF21B-KO MT dynamics, MTs are initialized with slightly different lengths, pulled from an exponential distribution with the mean of 1 μm. This was done so that even with very small catastrophe rates, MT lengths would not be equal. Simulations were run in a slightly altered version of *Cytosim* to be able to immobilize the centrosome in the center of the synapse (http://doi.org/10.5281/zenodo.4312942).

## Acknowledgements

This work was supported by the European Research Council Synergy grant 609822 and Netherlands Organization for Scientific Research ALW Open Program grant 824.15.017 to AA, as well as the European Research Council Consolidator Grant 819219 to LCK.

## Additional information

### Competing interests
Anna Akhmanova: Senior editor, *eLife*. The other authors declare that no competing interests exist.

### Funding

| Funder | Grant reference number | Author |
| --- | --- | --- |
| European Research Council | 609822 | Anna Akhmanova |
| European Research Council | 819219 | Lukas C Kapitein |
| Nederlandse Organisatie voor Wetenschappelijk Onderzoek | 824.15.017 | Anna Akhmanova |

The funders had no role in study design, data collection and interpretation, or the decision to submit the work for publication.

### Author contributions

Peter Jan Hooikaas, Hugo GJ Damstra, Conceptualization, Data curation, Formal analysis, Validation, Investigation, Visualization, Methodology, Writing - original draft, Writing - review and editing; Oane J Gros, Conceptualization, Software, Formal analysis, Investigation, Visualization, Methodology, Writing - original draft, Writing - review and editing; Wilhelmina E van Riel, Conceptualization, Resources, Formal analysis, Investigation, Methodology, Writing - review and editing; Maud Martin, Yesper TH Smits, Resources, Investigation; Jorg van Loosdregt, Resources, Methodology; Lukas C Kapitein, Conceptualization, Funding acquisition, Writing - original draft, Writing - review and editing; Florian Berger, Conceptualization, Software, Formal analysis, Supervision, Visualization, Writing - original draft, Project administration, Writing - review and editing; Anna Akhmanova, Conceptualization, Supervision, Funding acquisition, Writing - original draft, Project administration, Writing - review and editing

### Author ORCIDs

Peter Jan Hooikaas ![iD] https://orcid.org/0000-0001-9849-9193
Hugo GJ Damstra ![iD] https://orcid.org/0000-0003-0847-609X
Maud Martin ![iD] https://orcid.org/0000-0003-0048-6437
Lukas C Kapitein ![iD] http://orcid.org/0000-0001-9418-6739
Florian Berger ![iD] https://orcid.org/0000-0003-3355-4336
Anna Akhmanova ![iD] https://orcid.org/0000-0002-9048-8614

### Decision letter and Author response
Decision letter https://doi.org/10.7554/eLife.62876.sa1
Author response https://doi.org/10.7554/eLife.62876.sa2

## Additional files

### Supplementary files

• Transparent reporting form

### Data availability

Data generated or analysed during this study are included in the manuscript and supporting files.

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

# Appendix 1

**Appendix 1—key resources table**

| Reagent type (species) or resource | Designation | Source or reference | Identifiers | Additional information |
|---|---|---|---|---|
| Cell line (*Homo sapiens*) | HEK293T | ATCC | CRL-11268 | |
| Cell line (*Homo sapiens*) | Jurkat, Clone E6-1 | ATCC | TIB-152 | |
| Cell line (*Homo sapiens*) | Jurkat, KIF21B KO#1 | This paper | | CRISPR/Cas9 generated monoclonal Jurkat cell line |
| Cell line (*Homo sapiens*) | Jurkat, KIF21B KO#2 | This paper | | CRISPR/Cas9 generated monoclonal Jurkat cell line |
| Cell line (*Homo sapiens*) | Jurkat, KIF21B KO#1, re-expressing KIF21B-GFP | This paper | | Polyclonal line re-expressing KIF21B-GFP; generated from monoclonal KIF21B KO#1 Jurkat cells |
| Cell line (*Homo sapiens*) | Jurkat, KIF21B KO#2, re-expressing KIF21B-GFP | This paper | | Polyclonal line re-expressing KIF21B-GFP; generated from monoclonal KIF21B KO#2 Jurkat cells |
| Cell line (*Homo sapiens*) | Jurkat cells (control), expressing EB3-GFP | This paper | | Polyclonal Jurkat (control) line expressing EB3-GFP |
| Cell line (*Homo sapiens*) | Jurkat, KIF21B KO#1, expressing EB3-GFP | This paper | | Polyclonal line expressing EB3-GFP; generated from monoclonal KIF21B KO#1 Jurkat cells |
| Cell line (*Homo sapiens*) | Jurkat, KIF21B KO#2, expressing EB3-GFP | This paper | | Polyclonal line expressing EB3-GFP; generated from monoclonal KIF21B KO#2 Jurkat cells |
| Cell line (*Homo sapiens*) | Jurkat cells (control), expressing β-tubulin-GFP | This paper | | Polyclonal Jurkat (control) line expressing β-tubulin-GFP |
| Cell line (*Homo sapiens*) | Jurkat, KIF21B KO#1, expressing β-tubulin-GFP | This paper | | Polyclonal line expressing β-tubulin-GFP; generated from monoclonal KIF21B KO#1 Jurkat cells |
| Cell line (*Homo sapiens*) | Jurkat, KIF21B KO#2, expressing β-tubulin-GFP | This paper | | Polyclonal line expressing β-tubulin-GFP; generated from monoclonal KIF21B KO#2 Jurkat cells |
| Transfected construct (*Homo sapiens*) | KIF21B-GFP | This paper | | Lentiviral construct to transfect and express KIF21B-GFP in Jurkat cells |
| Transfected construct (*Homo sapiens*) | EB3-GFP | *Bouchet et al., 2016*; PMID:27939686 | | Lentiviral construct to transfect and express EB3-GFP in Jurkat cells |
| Transfected construct (*Homo sapiens*) | β-tubulin-GFP | *Bouchet et al., 2016*; PMID:27939686 | | Lentiviral construct to transfect and express β-tubulin-GFP in Jurkat cells |
| Transfected construct (*Homo sapiens*) | EB3-mCherry | *Stepanova et al., 2003*; PMID:12684451 | | Expression construct transfected in Jurkat cells |

*Continued on next page*

*Appendix 1—key resources table continued*

| Reagent type (species) or resource | Designation | Source or reference | Identifiers | Additional information |
|---|---|---|---|---|
| Sequence-based reagent | gRNA targeting sequence against KIF21B | This paper | gRNA sequence | caccgTGTGTGAGCAAGCTCATCGA |
| Sequence-based reagent | GAPDH_fw | This paper | qPCR primer | CAACGGATTTGGTCGTATT |
| Sequence-based reagent | GAPDH_rev | This paper | qPCR primer | GATGGCAACAATATCCACTT |
| Sequence-based reagent | IL-2_fw | This paper | qPCR primer | AACTCACCAGGATGCTCACATTTA |
| Sequence-based reagent | IL-2_rev | This paper | qPCR primer | TCCCTGGGTCTTAAGTGAAAGTTT |
| Antibody | Anti-CD3, clone UCHT1 (mouse monoclonal) | StemCell Technologies | Cat# #60011 | Coverslip coating (10 µg/mL) WB (1:400) |
| Antibody | Anti-HA, clone 16B12 (mouse monoclonal) | Biolegend (Covance) | Cat# MMS-101P, RRID: AB_10064068 | Coverslip coating (10 µg/mL) |
| Antibody | anti-KIF21B (rabbit polyclonal) | Sigma-Aldrich | Cat# HPA027249, RRID:AB_10602241 | WB (1:1000) |
| Antibody | Anti-GFP (rabbit polyclonal) | Abcam | Cat# Ab290, RRID:AB_303395 | WB (1:5000) |
| Antibody | Anti-Ku80 (mouse monoclonal) | BD Bioscience | Cat# 611360, RRID:AB_398882 | WB (1:2000) |
| Antibody | Anti- Lamtor4, clone D6A4V (rabbit monoclonal) | Cell Signalling Technology | Cat# 12284, RRID:AB_2797870 | IF (1:200) |
| Antibody | anti-CEP135 (rabbit polyclonal) | Sigma-Aldrich | Cat# SAB4503685; RRID:AB_10746232 | IF (1:200) |
| Antibody | Anti- α-tubulin, clone EP1332Y (rabbit monoclonal) | Abcam | Cat# ab52866, RRID:AB_869989 | IF (1:250) for ExM samples |
| Antibody | Anti-α-tubulin (mouse monoclonal) | Sigma-Aldrich | Cat# T6199, RRID:AB_477583 | IF (1:250) for STED samples WB (1:10000) |
| Antibody | Anti- α-tubulin, clone γL1/2 (rat monoclonal) | Abcam | Cat# Ab6160, RRID:AB_305328 | IF (1:300) |
| Antibody | Alexa Fluor 488-, 594- and 647-secondaries | Molecular Probes | | IF (1:200 – 1:400) |
| Antibody | IRDye 680LT and 800CW secondaries | Li-Cor Biosciences | | WB (1:10000) |
| Commercial assay or kit | Amaxa Cell Line Nucleofector kit V | Lonza | Cat# VPB-1002 | program X-001 or X-005 |
| Commercial assay or kit | iScript cDNA synthesis kit | Bio-Rad | Cat# 1708891 | |

*Continued on next page*

*Appendix 1—key resources table continued*

| Reagent type (species) or resource | Designation | Source or reference | Identifiers | Additional information |
|---|---|---|---|---|
| Commercial assay or kit | SYBR Select mastermix | Life Technologies | Cat# 44-729-19 | |
| Peptide, recombinant protein | Proteinase-K | Thermo Fisher | Cat# EO0491 | |
| peptide, recombinant protein | Monomeric GFP | This paper | | Obtained from HEK293T lysates containing overexpressed eGFP. (Clontech pEGFP-C1 vector) |
| Chemical compound, drug | acryloyl X-SE (AcX) | Thermo Fisher | Cat# A20770 | |
| Chemical compound, drug | sodium acrylate | Sigma-Aldrich | Cat# 408220 | |
| Chemical compound, drug | AA/BIS solution | Sigma-Aldrich | Cat# A3699 | |
| Chemical compound, drug | BIS | Sigma-Aldrich | Cat# M1533 | |
| Chemical compound, drug | cOmplete protease inhibitor cocktail | Roche | Cat# 4693132001 | |
| Chemical compound, drug | Puromycin | InvivoGen | Cat# ant-pr5b | (2 µg/mL) |
| Chemical compound, drug | Hygromycin | Invivogen | Cat# ant-hm | (100 µg/mL) |
| Chemical compound, drug | Polybrene | Merck-Millipore | Cat# TR-1003-G | (8 µg/mL) |
| Chemical compound, drug | Poly-D-Lysine | Thermo Fisher | Cat# A3890401 | |
| Chemical compound, drug | Vinblastine | Sigma-Aldrich | Cat# V1377 | |
| Chemical compound, drug | Phorbol 12-myristate 13-acetate (PMA) | Sigma-Aldrich | Cat# P8139 | |
| Chemical compound, drug | ionomycin | Sigma-Aldrich | Cat# I0634 | |
| Chemical compound, drug | TRIzol | Thermo Fisher Scientific | Cat# 15596026 | |
| Software, algorithm | GraphPad Prism | GraphPad Prism (https://graphpad.com) | RRID:SCR_015807 | |
| Software, algorithm | FIJI/ImageJ | FIJI/ImageJ (https://imagej.net/Fiji) | RRID:SCR_002285 | |

*Continued on next page*

*Appendix 1—key resources table continued*

| Reagent type (species) or resource | Designation | Source or reference | Identifiers | Additional information |
|---|---|---|---|---|
| Software, algorithm | ImageJ detection of molecules plugin (DoM) | *Chazeau et al., 2016*; PMID:26794511 | | |
| Software, algorithm | ImageJ KymoResliceWide plugin | | | https://github.com/ekatrukha/KymoResliceWide |
| Software, algorithm | ImageJ radiality plugin | *Martin et al., 2018*; PMID:29547120 | | https://github.com/ekatrukha/radialitymap |
| Software, algorithm | MetaMorph | Molecular Devices | RRID:SCR_002368 | |
| Software, algorithm | Leica Application Suite X | Leica Microsystems | RRID:SCR_013673 | |
| Software, algorithm | Micro-Manager | https://micro-manager.org/ | RRID:SCR_016865 | |
| Software, algorithm | Huygens Software | Scientific Volume Imaging https://svi.nl/HuygensSoftware | RRID:SCR_014237 | Drift correction of ExM sample acquisitions |
| Software, algorithm | Imaris, version 9.5.1 | Bitplane/Oxford instruments | RRID:SCR_007370 | |
| Software, algorithm | *Cytosim* | *Nedelec and Foethke, 2007*, PMID:19293826 | | |
| Software, algorithm | Python | https://www.python.org/ | RRID:SCR_008394 | |
| Software, algorithm | Seaborn | https://seaborn.pydata.org/ | RRID:SCR_018132 | |
| Software, algorithm | NumPy | https://numpy.org/ | RRID:SCR_008633 | |
| Software, algorithm | Pandas | https://pandas.pydata.org/ | RRID:SCR_018214 | |
| Software, algorithm | Adobe Illustrator | Adobe | RRID:SCR_010279 | Generation of cartoons and figures |
| Other | 8-well Chambered Coverglass w/ non-removable wells | Thermo | Cat# 155409 | |
| Other | Precision cover glasses thickness No. 1.5H | Marienfeld | Cat# 0107032 | Specific for ExM and STED samples |
| Other | silicone mold, 13mm inner diameter | Sigma-Aldrich | Cat# GBL664107 | |
| Other | Phalloidin-Alexa488 | Life Technologies | Cat# 12379 | IF (1:400) |
| Other | Phalloidin-Alexa594 | Life Technologies | Cat# 12381 | IF (1:400) |
| Other | DAPI-containing Vectashield mounting medium | Vector Laboratories | Cat# H-1200-10 | |
| Other | Vectashield mounting medium | Vector Laboratories | Cat# H-1000-10 | |
| Other | Prolong Gold | Thermo Fisher | Cat# P10144 | |

**Appendix 1—table 1.** Parameters used for simulations.

| Parameter | Value | Description/Reference |
|---|---|---|
| *MTs* | | |
| Polymerization speed | 0.3 µm/s | This study |
| Depolymerization speed | 1 µm/s | This study |
| Rigidity | 20 pN/µm | *Gittes et al., 1993* |
| Stall force | 5 pN | Describes the modulation of growth speed and catastrophe rate by antagonistic force (*Dogterom and Yurke, 1997*) |
| Catastrophe rate | 0.02 s$^{-1}$, 0.3 s$^{-1}$ | Matched to TIRF GFP-β-tubulin data (*Figure 4D-E*) |
| *Cell* | | |
| Viscosity | 0.1 pN.s/µm2 | Internal viscosity of T cells, like most blood cells, is usually estimated to be lower than that of somatic cells. Jurkat cell internal viscosity has been reported at different values (*Daza et al., 2019*; *Khakshour et al., 2015*). Because there is no consensus on the value, we used a viscosity such that the KIF21B-mediated polarization happens within the timescale measured for polarization. |
| Elasticity | 100 pN/µm | The spring stiffness of the cell for all objects with inertia. This same stiffness is used for interaction of MTs with the nucleus |
| Radius | 7 µm | This study, calculated from *Figure 1—figure supplement 2C-D*. |
| Synapse fraction | 0.9 | This corresponds to a synapse cutting off 10% of the height of the cell (1.4 µm). |
| Interpolation distance | 1 µm | The curvature starts 1 µm under the synapse (at 2.4 µm from the top of the unpolarized cell). |

*Continued on next page*

*Appendix 1—table 1 continued*

| Parameter | Value | Description/Reference |
|---|---|---|
| *centrosome* | | |
| First anchoring stiffness | 500 pN/μm | Rotational stiffness on the MTs at the center of the centrosome, as proposed previously (*Letort et al., 2016*). |
| Second anchoring stiffness | 500 pN/μm | Rotational stiffness on the MTs exerted at the periphery of the centrosome, as proposed previously (*Letort et al., 2016*). |
| Number of MTs | 90 | |
| *Dynein* | | |
| Walking speed | 1 μm/s | Average value from MT gliding over dynein (*Laan et al., 2012*) |
| Number | 50 | |
| Stall force | 4 pN | *Belyy et al., 2016* |
| Unbinding rate | 1 s⁻¹ | *Ohashi et al., 2019* |
| Initialization | uniform on synapse | Dynein is initialized on the synapse and part of the interpolated curve connecting the synapse to the rest of the cell. The region is defined as being within 1.6 μm of the synapse along the vertical axis. |
| Link stiffness | 100 pN/μm | Describes the elastic stiffness of the link between MT-binding site and anchoring point (*Letort et al., 2016*). |
| *KIF21B* | | |
| Walking speed | 0.71 μm/s | This study |
| MT state after a KIF21B-induced pause | Shrinkage | We inflate the effect of KIF21B to always cause catastrophe in order to more clearly display the effect of KIF21B as a delayed catastrophe inducer. |

*Continued on next page*

*Appendix 1—table 1 continued*

| Parameter | Value | Description/Reference |
|---|---|---|
| Capable of pausing a shrinking MT | No | This means that the KIF21B motors do not cooperatively pause: if one of them unbinds, the fiber is set to shrinkage and another bound KIF21B cannot pause this shrinking fiber. |
| *System* | | |
| Dimensionality | 2D | The overgrown KIF21B-KO MT system requires much computation per timestep, and thus we were not able to expand to 3D and keep our fit of the system. |
| Number of repeats | 30 per condition | In some figures, fewer repeats are shown for readability. If so, the ones were chosen that were run first chronologically, to avoid bias in run selection. |

