## [Decision Letter]

**Acceptance summary:**

This interesting study addresses the question of how the microtubule cytoskeleton reorganizes in the immunological synapse. Using a variety of experimental techniques, including expansion microscopy, and computer simulations, the work demonstrates an important role of microtubule length control by the kinesin-4 KIF21B for correct T cell polarization during immunological synapse formation, providing new insight into the molecular mechanism of this important process.

**Decision letter after peer review:**

Thank you for submitting your article "Kinesin-4 KIF21B limits microtubule growth to allow rapid centrosome polarization in T cells" for consideration by *eLife*. Your article has been reviewed by two peer reviewers, and the evaluation has been overseen by a Reviewing Editor and Vivek Malhotra as the Senior Editor. The following individuals involved in review of your submission have agreed to reveal their identity: Alex Mogilner (Reviewer #1); Michael L Dustin (Reviewer #2).

The reviewers have discussed the reviews with one another and the Reviewing Editor has drafted this decision to help you prepare a revised submission.

As the editors have judged that your manuscript is of interest, but as described below that additional experiments and modeling may be required before it is published, we would like to draw your attention to changes in our revision policy that we have made in response to COVID-19 (https://elifesciences.org/articles/57162). First, because many researchers have temporarily lost access to the labs, we will give authors as much time as they need to submit revised manuscripts. We are also offering, if you choose, to post the manuscript to bioRxiv (if it is not already there) along with this decision letter and a formal designation that the manuscript is "in revision at *eLife*". Please let us know if you would like to pursue this option. (If your work is more suitable for medRxiv, you will need to post the preprint yourself, as the mechanisms for us to do so are still in development.)

Summary:

This is a very interesting study addressing the question of microtubule cytoskeleton reorganization in the immunological synapse. Specifically, the work demonstrates the contribution of KIF21B for the control of the T cell microtubule (MT) network required for T cell polarization during immunological synapse formation. The authors use a variety of microscopy techniques, including expansion microscopy, controlled perturbations of the cell, and computer simulations to generate their results. The authors show that knockout of KIF21B results in longer MTs that result in an inability to polarise the MT network by a mechanism consistent with dynein motor function at the immunological synapse to capture long MTs and center the MT aster at the synapse. They use the Jurkat cell line, which is a classical model for this step in immune synapse function and fully appropriate. They show that KIF21B-GFP can rescue the knockout phenotype and then use this as a way to follow KIF12B dynamics in the Jurkat cells. KIF21B works by inducing pausing and catastrophe, thus, more MTs are shorter when present. They also rescue the defect in the KIF21B KOs with 0.5 nM vinblastine, that directly increases catastrophes, shortens the MTs and restores MT network polarization to the synapse. As a functional surrogate they investigate lysosome positioning at the synapse, which is one of the proposed functions of this cytoskeletal polarization. The use of expansion microscopy in this system is relatively new and clearly very powerful. The modelling component adds to the story and supports the sliding model proposed by Poenie and colleagues in 2006, but cannot say that there is no component of end capture and shrinkage as proposed by Hammer and colleagues more recently. Experiments and modelling are performed to a high standard and the results advance the field.

Essential revisions:

1) The author use poly-D-lysine (PDL) to attach the Jurkat cells to the coverslip as a "control" condition. Do the authors also observe polarization of the MT cytoskeleton in these experiments? The intention is probably to have a random orientation, but it has been found that charge-based adhesion zones can activate T cells due to differential exclusion of CD45 and TCR or other effects (see PMID: 29476188). Please always state when PDL + anti-CD3 is used PDL or was a control antibody used? This should be stated in each case as its not necessarily neutral for these sensitive cells. If the authors have data on polarization on PDL they should report this in supplementary figures. If there is potential activation in this condition based on some non-random polarization then it would be useful to briefly discuss this as the questions being asked don't really rely on the cells being "resting" or "activated", although things like Ca^2+^ flux might affect the function of KIF21B.

2) The authors use a mathematical model and this theoretical part was well performed. The authors have done a good job of discussing the earlier work from Poenie with polarization microscopy that favoured the sliding model and the recent work from Hammer that suggested the end capture/shrinkage model might better fit direct observations, but neither study provided relevant perturbations to test the models. Another recent modelling study in Biophys J seems to support a mix of the two mechanisms being relevant – PMID: 31084903. It seems intuitive that the sliding model works if all the circumferential MTs are <πr in length but becomes problematic when MTs are in the πr->2πr length range, where a "tug of war" can happen. It seems like there are lots of MTs >πr length in the KIF21B KO based on data in Figure 2G. Probably this has been corrected for the expansion and relates to lengths in the native cells. The model doesn't seem to fully capture this as even without any KIF21B the length is 12 µm- which is similar to wildtype/no perturbation rather than the estimated values for the KIR21B KO of ~ 30 µm. What happens if parameters are adjusted to allow longer MTs to form in the model in both WT and KIF21K KO setting? This could be explored by additional simulations or by a discussion if deemed beyond scope.

3) The model nicely integrates and explains the data, but is it predictive? A detailed model like the one here clearly can generate some nontrivial prediction that could either be experimentally tested here or proposed to be tested in the future.

---

## [Author Response]

Essential revisions:1) The author use poly-D-lysine (PDL) to attach the Jurkat cells to the coverslip as a "control" condition. Do the authors also observe polarization of the MT cytoskeleton in these experiments? The intention is probably to have a random orientation, but it has been found that charge-based adhesion zones can activate T cells due to differential exclusion of CD45 and TCR or other effects (see PMID: 29476188). Please always state when PDL + anti-CD3 is used PDL or was a control antibody used? This should be stated in each case as its not necessarily neutral for these sensitive cells. If the authors have data on polarization on PDL they should report this in supplementary figures. If there is potential activation in this condition based on some non-random polarization then it would be useful to briefly discuss this as the questions being asked don't really rely on the cells being "resting" or "activated", although things like Ca^2+^ flux might affect the function of KIF21B.

As the reviewer pointed out, we made use of poly-D-lysine (PDL) to improve adherence of Jurkat cells to coverslips when performing immunofluorescence experiments and, in addition, used PDL to image non-activated Jurkat cells in Figure 1H. To clarify that in all other figure panels involving immunofluorescence techniques we made use of PDL-coating + anti-CD3 we have added this information to all relevant figure legends. In addition, we have performed immunofluorescence experiments to quantify centrosome polarization on three different surface conditions: “PDL only”, “PDL + anti-CD3” and “PDL + anti-HA (control antibody)”. In this experiment, we found that whereas the combination of PDL and anti-CD3 potently induced centrosome polarization in Jurkat cells, neither PDL alone nor PDL + control antibody were sufficient to trigger such an effect. The results of this experiment are included in the new Figure 1—figure supplement 2H-I. We think that the 27.4% (PDL only) and 24.9% (PDL + anti-HA) of cells with the centrosome visible in the TIRF plane correspond to random orientations of cells attached to the glass surface. Since these cells were fixed with methanol, which compacts cells in the z-dimension, the percentage of cells with centrosomes close to de imaging plane would likely have been even smaller before fixation. In contrast, the experimental condition that included an anti-CD3 antibody showed robust polarization with 79.3 % of cells having a centrosome visible at the imaging plane.

2) The authors use a mathematical model and this theoretical part was well performed. The authors have done a good job of discussing the earlier work from Poenie with polarization microscopy that favoured the sliding model and the recent work from Hammer that suggested the end capture/shrinkage model might better fit direct observations, but neither study provided relevant perturbations to test the models. Another recent modelling study in Biophys J seems to support a mix of the two mechanisms being relevant – PMID: 31084903. It seems intuitive that the sliding model works if all the circumferential MTs are <πr in length but becomes problematic when MTs are in the πr->2πr length range, where a "tug of war" can happen. It seems like there are lots of MTs >πr length in the KIF21B KO based on data in Figure 2G. Probably this has been corrected for the expansion and relates to lengths in the native cells. The model doesn't seem to fully capture this as even without any KIF21B the length is 12 µm- which is similar to wildtype/no perturbation rather than the estimated values for the KIR21B KO of ~ 30 µm. What happens if parameters are adjusted to allow longer MTs to form in the model in both WT and KIF21K KO setting? This could be explored by additional simulations or by a discussion if deemed beyond scope.

We agree with the reviewer that intuitively one would expect that the fraction of MTs longer than half the circumference of the cell would indicate if centrosome repositioning could be successful. However, the connection between MT length and MT capture by dynein at the synapse cannot be reduced purely to MT length. The relative positions of the centrosome and nucleus, together with the centrosomal angle of a single MT define how the MT conforms to the space, and thus defines how likely the capture is. A fiber directing straight towards the synapse needs a much shorter MT length for capture than a fiber curving under the plasma membrane and surrounding the nucleus. These individual properties are all integrated into the measured values of capture, which we report in Figure 6F-H. Furthermore, as we discuss in the fifth paragraph of the Discussion, fluctuations are important to break the symmetry to allow microtubules to connect to the synapse along only one side of the nucleus. This mechanism is not well described by intuitive argument to compare the average length of the MTs to the circumference of the cell.

As pointed out by the reviewer, we did not match the MT lengths of our simulations to the lengths determined in our experiments for two reasons. First, we found it problematic to use the MT lengths from a three-dimensional system for a two-dimensional simulation. Obviously, in a 2D system with long MTs, estimated from a 3D system, the ratio of MT occupied space to empty space is drastically changed. The second reason is that such a distribution of lengths will be exponential. The long fibers in the tail of the exponential distribution with about ~30 µm length are very difficult to compute. We ran a few simulations on a high-performance computer cluster, but the numerical solution per time step (0.04 modeled seconds) increased to hundreds of seconds, which resulted in lengthy simulations which we terminated after 30 hours, out of concerns over computing time. Further investigations into these systems are beyond our possibilities and resources. In conclusion, we believe that using the measurements of the dynamics of MT to calibrate our simulations is the most internally consistent way to model the system. To fully clarify this important point raised by the reviewer, we added the following text to the Discussion:

“Because of this dimensionality reduction, it is not possible to quantitatively compare all experimental results to the results of our simulations. […] Therefore, our approach to calibrate the model to the measured MT dynamics seems the most reasonable.”

3) The model nicely integrates and explains the data, but is it predictive? A detailed model like the one here clearly can generate some nontrivial prediction that could either be experimentally tested here or proposed to be tested in the future.

As recognized by the reviewer, the main focus of our model was to “integrate and explain the data”. Nonetheless, we can draw at least two nontrivial predictions from the model. A strong prediction with important consequences is the length regulation of MTs by only a small number of KIF21B molecules. This length regulation mechanism could be tested in a reconstituted in vitro system in which the dependence on the number of KIF21B molecules can be systematically changed, or by exact quantification of KIF21B units through fluorescent labeling. This prediction could also potentially be tested in vivo, by the rescue of KIF21B knockout with KIF21B-GFP at different expression levels. However, these experimental validations of the small number of involved KIF21B molecules are very laborious and beyond the scope of this study. The second prediction is related to the KIF21B knockout system. In such a system the centrosome is not repositioned to the synapse. Our simulations suggest that in this case, the MT network is under constant force, but not able to rearrange. Therefore, we predict strong deformations of the nucleus by the MT network. However, we did not directly investigate such deformations in our simulations in which the nucleus is a rather stiff object. To emphasize the predictions from our model, we added the following text in the fourth paragraph of the Discussion:

“Our simulations predict that a relatively small number of KIF21B molecules is sufficient to restrict MT length. […] This length regulation by only a small number of KIF21B molecules could be validated experimentally by different labeling strategies or in a reconstituted system in vitro.”

We also added a text in the sixth paragraph of the Discussion:

“Our simulations suggest that when centrosome translocation is impaired, the MT network is experiencing balanced forces. […] It could be also possible that nuclear deformations push MTs towards the synapse, where they form dense peripheral MT bundles to accommodate the least curvature (Figure 2A and B).”